# Learning from Interval Targets

**Rattana Pukdee** *
Carnegie Mellon University
rpukdee@cs.cmu.edu

**Ziqi Ke**
Bloomberg
zke7@bloomberg.net

**Chirag Gupta**
Bloomberg
cgupta61@bloomberg.net

## Abstract

We study the problem of regression with interval targets, where only upper and lower bounds on target values are available in the form of intervals. This problem arises when the exact target label is expensive or impossible to obtain, due to inherent uncertainties. In the absence of exact targets, traditional regression loss functions cannot be used. First, we study the methodology of using a loss function compatible with interval targets, for which we establish non-asymptotic generalization bounds based on smoothness of the hypothesis class that significantly relax prior assumptions. Second, we propose a novel minmax learning formulation: *minimize* against the worst-case (*maximized*) target labels within the provided intervals. The maximization problem in the latter is non-convex, but we show that good performance can be achieved by incorporating smoothness constraints. Finally, we perform extensive experiments on real-world datasets and show that our methods achieve state-of-the-art performance.

## 1 Introduction

Supervised learning has achieved significant empirical success, largely due to the availability of extensive labeled datasets. However, in many real-world tasks, obtaining target labels is challenging, which hampers the performance of these methods. This difficulty arises either from high labeling costs—for example, certain medical measurements are expensive—or from practical limitations, such as sensors that only record target values at discrete intervals (e.g., every hour), leaving intermediate values unobserved. Prior work has addressed this issue by incorporating additional information into the learning pipeline. For instance, some approaches encourage model outputs to be smooth over unlabeled data [Zhu, 2005, Chapelle et al.], while others enforce models to satisfy constraints derived from domain knowledge, such as physical laws [Willard et al., 2020, Swischuk et al., 2019].

In this work, we focus on regression tasks where only the lower and upper bounds of the target values (intervals) are available. Our setting relates to both weak supervision and learning with side information. Learning with interval targets generalizes supervised learning, which corresponds to the special case where the lower and upper bounds are equal. On the other hand, for many tasks, it is easier and more practical for human labelers to provide interval targets instead of precise single values; thus, these intervals can be viewed as a form of weak supervision. Additionally, in various settings, such intervals are readily available for unlabeled data, either from domain knowledge or inherent properties of the data, serving as side information e.g., in bond pricing.

A natural strategy for learning from interval targets is to learn a hypothesis whose outputs always lie within the provided intervals. Despite its simplicity, previous work [Cheng et al., 2023a] has shown that this method leads to a hypothesis that converges to the optimal one under two assumptions: (i) the true target function belongs to the hypothesis class, and (ii) the intervals have an ambiguity degree

---

*This work was conducted during an internship at Bloomberg.

39th Conference on Neural Information Processing Systems (NeurIPS 2025).

smaller than 1 (Section 2). However, these assumptions are unlikely to hold in practice. In particular, (ii) is often violated; for example, even in the simple case where the interval is a ball of radius $\epsilon$ around the target value $y$, the ambiguity degree equals 1. It is important to understand whether this approach can be effective under more relaxed assumptions.

## 1.1 Summary of contributions

- First, we study the approach of modifying the typical regression loss to make it compatible with interval learning. This setup was first studied by Cheng et al. [2023a], and our result improves upon theirs. We show that for any hypothesis class $\mathcal{F}$ with Rademacher complexity decaying as $O(1/\sqrt{n})$ such as for a class of two-layer neural networks with bounded weights, we prove that, with high probability, the error decomposes into an irreducible term depending on the quality of the intervals and the Lipschitz constant of the hypothesis class, plus terms that vanish at $O(1/\sqrt{n})$ (Theorem 4.1). Compared to the previous bound by [Cheng et al., 2023a], our result: (1) applies even when a so-called "ambiguity degree" is large (this roughly corresponds to going from the well-specified case to the agnostic case), (2) provides non-asymptotic guarantees, and (3) reveals how hypothesis class structure affects the learning guarantee. The key insight is that, when the hypothesis class is smooth, the outputs for two close inputs cannot differ significantly. As a result, portions of the original intervals can be ruled out, leading to much smaller valid intervals (Theorem 3.6 and Figure 2).

- Second, we explore an alternative approach that learns a hypothesis minimizing the loss with respect to the worst-case labels within the given intervals. Since we assume that the true target values lie within these intervals, the worst-case loss serves as an upper bound on the regression loss. We consider two variants of the second approach: i) we allow the worst-case labels to be any points within the intervals, ii) we restrict the worst-case labels to be outputs of some hypothesis in our hypothesis class, thereby incorporating the smoothness property. We show that there are scenarios where the second variant performs arbitrarily better than the first (Proposition 5.4), indicating that constraining the worst-case labels to the hypothesis class is preferable in the worst-case scenario.

- We complement the theory with experiments that demonstrate the effectiveness of both methods on real-world datasets.

## 1.2 Related work

Our problem is closely related to partial-label learning, where each training point is associated with a set of candidate labels instead of a single target label [Cour et al., 2011, Ishida et al., 2017, Feng et al., 2020a, Ishida et al., 2019, Yu et al., 2018]. In classification with finite label sets, common approaches include minimizing the average loss over the label set [Jin and Ghahramani, 2002, Zhang et al., 2017, Wang et al., 2019, Xu et al., 2021, Wu et al., 2022, Gong et al., 2022] and identifying the true label from the candidate set [Lv et al., 2020, Zhang et al., 2016, Yu and Zhang, 2016]. Theoretical work has established learnability conditions [Liu and Dietterich, 2014, Cour et al., 2011] and statistically consistent estimators [Lv et al., 2020, Feng et al., 2020b, Wen et al., 2021] based on the small ambiguity degree assumption or specific label set generating distributions.

The regression setting has received less attention. While Cheng et al. [2023b] introduced partial-label regression with finite label sets and Cheng et al. [2023a] extended it to intervals, both rely heavily on the small ambiguity degree assumption. However, this assumption—originally proposed for classification Cour et al. [2011]—may not be suitable for regression tasks. In classification, a hypothesis is either correct or incorrect, and a small ambiguity degree ensures that, with enough observed label sets, we can recover the true label. However, in regression, we are often satisfied with predictions that are sufficiently close to the target—for example, within an error tolerance of $\epsilon$—making the concept of ambiguity degree less applicable. We explore a natural extension of the ambiguity degree to ambiguity radius for the regression task in Section F and argue that our theoretical analysis not only is applicable to this extension but do also provide a stronger result. In our work, we study a projection loss, which is equivalent to the partial-label learning loss (PLL loss) in Lv et al. [2020] for the classification, and generalizing the limiting method in Cheng et al. [2023a]. We provide a non-asymptotic error bound that does not rely on the ambiguity degree and extend our analysis to the agnostic setting. Additional related work appears in Appendix A.

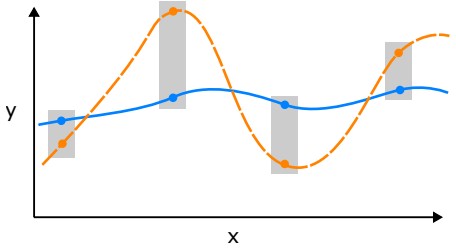
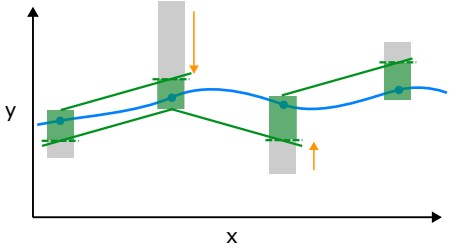

| Figure 1: One dimension example of learning from interval targets | Figure 2: Smooth hypothesis leads to a smaller interval |

Figure 3: (1) An example of learning from intervals where the input is one dimension. The intervals are shown as gray boxes. A natural method is to learn a hypothesis that always lies within these intervals. Here, we illustrate two such hypotheses that are both valid but have different levels of smoothness. (2) When the hypothesis is smooth (blue line), it lies within intervals much smaller than the original ones, depicted by the green region (Proposition 3.4). We can extend this result to hypotheses that approximately lie within the intervals (Theorem 3.6).

## 1.3 Preliminaries and notation

Let $\mathcal{X}$ be the feature space and $\mathcal{Y}$ be the label space. Let $f^*\colon \mathcal{X} \to \mathcal{Y}$ denote the target function. We use uppercase letters (e.g., $X$) to represent random variables and lowercase letters (e.g., $x$) for deterministic variables. We consider a regression problem where our goal is to learn a function $f\colon \mathcal{X} \to \mathcal{Y}$ from a hypothesis class $\mathcal{F}$ that approximates the target function $f^*$ in the deterministic label setting. Let $\mathcal{D}$ be the distribution over $\mathcal{X} \times \mathcal{Y}$ where, for each $x \in \mathcal{X}$, the label $y$ is deterministically given by $y = f^*(x)$. **Our goal is to learn a function $f$ that minimizes the expected loss** $\mathrm{err}(f) := \mathbb{E}_{(X,Y)\sim\mathcal{D}}\big[\ell\big(f(X),Y\big)\big]$ **for some loss function** $\ell\colon \mathcal{Y} \times \mathcal{Y} \to \mathbb{R}$, satisfying the following,

**Assumption 1.** *The loss function $\ell : \mathcal{Y} \times \mathcal{Y} \to \mathbb{R}$ can be written as $\ell(y,y') = \psi(|y-y'|)$ for some non-decreasing function $\psi$, and satisfies $\ell(y,y') = 0$ if and only if $y = y'$.*

**Interval targets.** We assume that we have access only to interval samples of the form $\{(x_i,l_i,u_i)\}_{i=1}^n$, where $l_i$ and $u_i$ are the lower and upper bounds of $y_i$, respectively. While we assume that the label is fixed to $f^*(x_i)$, we allow the intervals—that is, the bounds $(l_i,u_i)$—to be random and assume that each tuple $(x_i,l_i,u_i)$ is sampled from some distribution $\mathcal{D}_I$. To deal with singular events of measure zero, we assume that $\mathcal{D}_I$ is a nonatomic distribution i.e. it does not contain a point mass (see Appendix D for a full definition). We also use $p$ to refer to the probability density function.

## 2 Learning from intervals using a projection loss

Since the target label $y$ always lies within the interval $[l,u]$, a natural strategy is to learn a hypothesis $f \in \mathcal{F}$ such that $f(x) \in [l,u]$ for all $x \in \mathcal{X}$ (Figure 1). In previous work, Cheng et al. [2023a] analyzed the following strategy.

$$\text{Learn } f \text{ that minimizes the empirical risk of the 0-1 loss: } \sum_{i=1}^n \ell_{0-1}(f(x_i),l_i,u_i), \qquad (1)$$

where $\ell_{0-1}(f(x),l,u) := 1[f(x) < l] + 1[f(x) > u]$. Using $\ell_1$ loss as the surrogate (equation (12)), they showed that $f$ converges to $f^*$ as $n \to \infty$ if two assumptions are satisfied, (i) Realizability, that is, $f^* \in \mathcal{F}$, (ii) Ambiguity degree is smaller than 1. Ambiguity degree is the maximum probability of a specific incorrect target $y'$, belonging to the same interval $[l,u]$ as the true target $y$:

$$\text{Ambiguity degree}(\mathcal{D},\mathcal{D}_I) := \sup_{(x,y,y')} \left\{ \Pr_{\mathcal{D}_\mathcal{I}}(y' \in [L,U] \mid X = x) : p_{\mathcal{D}}(x,y) > 0,\ y' \neq y \right\} < 1 \qquad (2)$$

These assumptions can be impractical and restrictive. First, our hypothesis class may not contain $f^*$. Second, an ambiguity degree smaller than 1 implies that for any fixed $x$, if we keep sampling the interval $[l,u]$, the intersection of such intervals (in the limit) would only be the set of the true

target $\{y\}$; that is, we can recover the true $y$ given an infinite number of intervals. However, this assumption is unlikely to hold in practice because there is usually a gap between the upper and lower bounds and the target $y$. For example, in the simple case where $[l, u] = [y - \epsilon, y + \epsilon]$ (a ball with radius $\epsilon > 0$ around the true target $y$), the assumption fails since $y + \epsilon/2$ always lies within the interval at the same time with the true $y$.

We begin by defining a suitable learning objective. Since the 0-1 loss above is not continuous, it is not suitable for gradient-based optimization techniques. To address this, we relax the loss by considering a projection

$$\pi_\ell(f(x), l, u) := \min_{\tilde{y} \in [l, u]} \ell(f(x), \tilde{y}) \tag{3}$$

for any general loss function $\ell$. The following proposition shows that $\pi_\ell$ is a meaningful proxy for the 0-1 loss, and can be evaluated efficiently by only considering the boundaries of the interval.

**Proposition 2.1.** *Suppose that $\ell : \mathcal{Y} \times \mathcal{Y} \to \mathbb{R}$ is a loss function that satisfies Assumption 1 then $\pi_\ell(f(x), l, u) = 0$ if and only if $f(x) \in [l, u]$, and we can write*

$$\pi_\ell(f(x), l, u) = 1[f(x) < l]\ell(f(x), l) + 1[f(x) > u]\ell(f(x), u). \tag{4}$$

The proof is provided in Appendix C.1. In the rest of the paper, we refer to $\pi_l$ as the **projection loss**. Consequently, the informal goal given in equation 1 can be formalized as the following objective:

$$\min_f \sum_{i=1}^n 1[f(x_i) < l_i]\ell(f(x_i), l_i) + 1[f(x_i) > u_i]\ell(f(x_i), u_i). \tag{5}$$

## 3  Properties of a hypothesis that lie inside the interval targets

We will derive key properties of a hypothesis that lie inside the interval targets which will provide an essential setup for our main theoretical results in the next section. We denote $\widetilde{\mathcal{F}}_\eta := \{f \in \mathcal{F} \mid \mathbb{E}[\pi_\ell(f(X), L, U)] \leq \eta\}$ as a class of hypotheses with the expected projection loss is smaller than $\eta$. This is an interesting hypothesis class to study because as we minimize the projection objective equation 5, a uniform convergence argument (e.g. Mohri [2018]) would guarantee that the result hypothesis $f$ belong to $\widetilde{\mathcal{F}}_\eta$. The value of $\eta$ depends on the number of data points and the complexity of $\mathcal{F}$. In particular, with probability at least $1 - \delta$ over the draws $(x_i, l_i, u_i) \sim \mathcal{D}_I$, for all, $f \in \mathcal{F}$,

$$\mathbb{E}[\pi_\ell(f(X), L, U)] \leq \frac{1}{n} \sum_{i=1}^n \pi_\ell(f(x_i), l_i, u_i) + 2R_n(\Pi(\mathcal{F})) + M\sqrt{\frac{\ln(1/\delta)}{n}}. \tag{6}$$

Here, $R_n(\Pi(\mathcal{F}))$ is the Rademacher complexity of the function class $\Pi(\mathcal{F}) := \{\pi_\ell(f(x), l, u) \mapsto \mathbb{R} \mid f \in \mathcal{F}\}$ and we assume that the $\pi_\ell$ is uniformly bounded by $M$. Thus, given $n$, $M$, and the empirical loss on observed data (first term in R.H.S.), we have an **upper bound** of $\eta$ which $f \in \widetilde{\mathcal{F}}_\eta$ which decreases with $n$. In the rest of this section, we will provide a property of a hypothesis $f \in \widetilde{\mathcal{F}}_\eta$ for any fixed $\eta > 0$. In particular, we show that for any $x$, $f(x)$ belongs to an interval that is smaller than the original interval targets (Theorem 3.6) where the size of the reduced intervals depend on the Lipschitz constant of $\mathcal{F}$ and $\eta$. This leads to our main result: a generalization bound on the loss of $f$ w.r.t. actual labels $y$, thus showing that regression can be done using interval targets (Section 4).

### 3.1  Effect of realizability and small ambiguity degree assumptions on $\widetilde{\mathcal{F}}_\eta$

We begin by examining the implications of the assumptions made in prior work (Section 2). The realizability assumption implies that $f^* \in \widetilde{\mathcal{F}}_0$ since the projection loss of $f^*$ is always zero. Second, the small ambiguity degree assumption implies that, for any $x$, the intersection of the intervals can only be the singleton set $\{y\}$. As a result, we have $\widetilde{\mathcal{F}}_0 = \{f \in \mathcal{F} \mid \text{err}(f) = 0\} \neq \emptyset$.

With these assumptions, we can show that minimizing the projection objective will converge to a hypothesis with zero error. The following informal argument summarizes the asymptotic analysis

of Cheng et al. [2023b]. Here is the high-level idea: let $f_n$ be the hypothesis that minimizes the empirical projection objective equation 5. Realizability implies that there exists $f^* \in \mathcal{F}$ with an expected loss of zero. Since $f_n$ achieves the empirical risk no larger than that of $f^*$, it must achieve an empirical risk of zero. From equation 6, we have $f_n \in \widetilde{\mathcal{F}}_{\eta_n}$ with high probability, where $\eta_n = 2R_n(\Pi(\mathcal{F})) + M\sqrt{\frac{\ln(1/\delta)}{n}}$. In general, for a hypothesis class with the Rademacher complexity decays as $O(1/\sqrt{n})$, we have $\eta_n = O(1/\sqrt{n})$. Now as $n \to \infty$, we have $\eta_n \to 0$ which means that $\widetilde{\mathcal{F}}_{\eta_n} \to \widetilde{\mathcal{F}}_0$. Consequently, $\text{err}(f_n) \to 0$ since any member of $\widetilde{\mathcal{F}}_0$ has zero error.

However, when the realizability and ambiguity degree assumptions do not hold, there may be $f \in \widetilde{\mathcal{F}}_0$ with $\text{err}(f) > 0$. Additionally, with a finite amount of data, we can only learn a hypothesis $f \in \widetilde{\mathcal{F}}_\eta$ for some $\eta > 0$. In the next section, we will analyze $\widetilde{\mathcal{F}}_\eta$ without relying on the small ambiguity degree assumption and in finite samples.

## 3.2 Properties of $\widetilde{\mathcal{F}}_\eta$

Although our results extend to the probabilistic interval setting, where multiple intervals $[l, u]$ are drawn for each $x$, we focus on the deterministic interval setting in the main paper for simplicity. In this case, each $x$ is associated with a fixed interval $[l_x, u_x]$. A detailed discussion of the probabilistic interval setting is in Appendix D. Now, we start with the following characterization of $f(x)$ for $f \in \widetilde{\mathcal{F}}_0$ and then later we will consider when $f \in \widetilde{\mathcal{F}}_\eta$. First, we can see that when the expected projection loss is zero, $f(x)$ must lie inside the given interval.

**Proposition 3.1.** *For any $f \in \widetilde{\mathcal{F}}_0$, we have $f(x) \in [l_x, u_x]$ for any $x$ with $p(x) > 0$.*

The proof is based on the Assumption 2 and the fact that the expected projection loss is zero. Next, we can further show that the interval in which $f(x)$ must lie can be made smaller than $[l_x, u_x]$) if we assume that the class $\mathcal{F}$ contains only $m$-Lipschitz function.

**Definition 3.2** (m-Lipschitz). *A class $\mathcal{F}$ is $m$-Lipschitz when for any $f \in \mathcal{F}$ and any $x, x' \in \mathcal{X}$*

$$|f(x) - f(x')| \le m\|x - x'\| \tag{7}$$

We can rearrange the inequality into $f(x') - m\|x - x'\| \le f(x) \le f(x') + m\|x - x'\|$. For $f \in \widetilde{\mathcal{F}}_0$, we can substitute $f(x')$ with its lower and upper bound $l_{x'}, u_{x'}$, which implies $l_{x'} - m\|x - x'\| \le f(x) \le u_{x'} + m\|x - x'\|$. We denote this as a lower and upper bound of $f(x)$ induced by $x'$.

**Definition 3.3** (A lower and upper bound induced by $x'$). *For any $x, x' \in \mathcal{X}$, a lower and upper bound of $f(x)$ induced by $x'$ is given by*

$$l_{x' \to x}^{(m)} := l_{x'} - m\|x - x'\|, u_{x' \to x}^{(m)} := u_{x'} + m\|x - x'\|.$$

*Furthermore, the intersection of such bound over all $x'$ with $p(x') > 0$ is denoted by*

$$[l_{\mathcal{D} \to x}^{(m)}, u_{\mathcal{D} \to x}^{(m)}] = \bigcap_{p(x') > 0} [l_{x' \to x}^{(m)}, u_{x' \to x}^{(m)}]. \tag{8}$$

Following the argument above, we can derive a reduced interval for any $f \in \widetilde{\mathcal{F}}_0$.

**Proposition 3.4.** *Let $\mathcal{F}$ be a class of hypotheses that are $m$-Lipschitz and suppose that $\ell$ satisfies Assumption 1. Then for any $f \in \widetilde{\mathcal{F}}_0$ and for each $x$ with $p(x) > 0$,*

$$f(x) \in [l_{\mathcal{D} \to x}^{(m)}, u_{\mathcal{D} \to x}^{(m)}]. \tag{9}$$

First, we observe that $[l_{\mathcal{D} \to x}^{(m)}, u_{\mathcal{D} \to x}^{(m)}]$ is always smaller than $[l_x, u_x]$ because when we set $x' = x$, we have $[l_{x' \to x}^{(m)}, u_{x' \to x}^{(m)}] = [l_x, u_x]$. Second, if the hypothesis becomes more smooth, the interval $[l_{\mathcal{D} \to x}^{(m)}, u_{\mathcal{D} \to x}^{(m)}]$ gets smaller. This phenomenon can also be interpreted as implicitly "denoising" the original intervals by leveraging the smoothness of the hypothesis class.

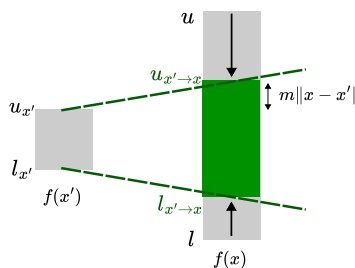

Figure 4: An interval of $x$ induced by $x'$

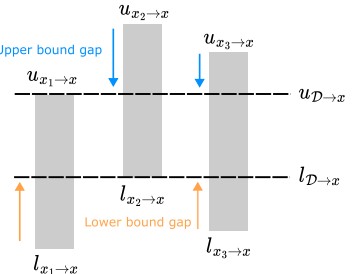

Figure 5: Upper and lower bound gaps

Figure 6: (4) Based on the smoothness property, the difference between $f(x)$ and $f(x')$ cannot exceed $m\|x - x'\|$. As a result, the upper and lower bounds of $f(x')$ imply the corresponding bounds for $f(x)$. (5) The lower bound gap of $x'$ to $x$ is defined as the difference between the lower bound of $f(x)$ induced by $x'$ and the largest lower bound ($\tilde{l}_{\mathcal{D} \to x}^{(m)}$); similarly for the upper bound gap. These gaps are crucial in bounding the size of $r_\eta(x)$ and $s_\eta(x)$ (how much we have to compensate when $f \in \widetilde{\mathcal{F}}_\eta$) where larger gaps lead to larger values (Theorem 3.6).

Next, we extend Proposition 3.4 to $\widetilde{\mathcal{F}}_\eta$. The key technical challenge is that for $f \in \widetilde{\mathcal{F}}_\eta$, $f(x)$ may lie outside the interval so we can't simply use $l_{x'}, u_{x'}$ as lower and upper bounds of $f(x')$ anymore. This complicates the application of the Lipschitz property because $f(x')$ can now be arbitrarily large or small for any $x'$, as long as the expected projection loss is smaller than $\eta$. The following result uses a new notion of a bound gap of $f(x)$ induced by $x'$ which is the difference between the lower and upper bounds induced by a given $x'$ and the best lower and upper bounds from all $x'$ (Figure 5).

**Definition 3.5** (A lower and upper bound gap induced by $x'$). *We uses the notation,*

$$lg_{x' \to x}^{(m)} = l_{\mathcal{D} \to x} - l_{x' \to x}^{(m)}, \quad \text{and} \quad ug_{x' \to x}^{(m)} = u_{x' \to x}^{(m)} - u_{\mathcal{D} \to x},$$

*to respectively denote the lower bound gap and upper bound gap for $f(x)$ induced by $x'$.*

**Theorem 3.6.** *Let $\mathcal{F}$ be a class of functions that are $m$-Lipschitz, and $\ell(y, y') = |y - y'|^p$ for any $p \geq 1$. For any $f \in \widetilde{\mathcal{F}}_\eta$ and for each $x$ with $p(x) > 0$ we have,*

$$f(x) \in [l_{\mathcal{D} \to x}^{(m)} - r_\eta(x), u_{\mathcal{D} \to x}^{(m)} + s_\eta(x)], \text{ where,} \tag{10}$$

$$r_\eta(x) = r \quad s.t. \quad \mathbb{E}_X[(r - lg_{X \to x}^{(m)})_+^p] = \eta, and \tag{11}$$

$$s_\eta(x) = s \quad s.t. \quad \mathbb{E}_X[(s - ug_{X \to x}^{(m)})_+^p] = \eta. \tag{12}$$

*Proof.* (Sketch) The proof leverages the smoothness property of $f$ to establish bounds on how far the function values can deviate from their projected intervals. The key insight is that if $f(x)$ significantly deviates from the reduced interval $[l_{\mathcal{D} \to x}^{(m)}, u_{\mathcal{D} \to x}^{(m)}]$, then by Lipschitz continuity, $f(x')$ must also deviate from $[l_{x'}, u_{x'}]$ for nearby points $x'$. However, such deviations are constrained by the expected projection loss being bounded by $\eta$. The proof proceeds in three main steps: i) using the Lipschitz property, we show that if $f(x)$ deviates below its lower bound $l_{\mathcal{D} \to x}^{(m)}$ by some amount $r$, then for all points $x'$: $f(x') \leq \tilde{l}_{x'} - (r - (\tilde{l}_{\mathcal{D} \to x}^{(m)} - \tilde{l}_{x \to x}^{(m)}))$, ii) the projection loss bound $\mathbb{E}[\pi_\ell(f(X), L, U)] \leq \eta$ implies that such deviations cannot be too large. iii) the maximum possible deviation $r_\eta(x)$ is characterized by the equation: $\eta = \mathbb{E}[1[g(x, X, r) < L]\ell(g(x, X, r), L)]$ where $g(x, x', r)$ represents the upper bound on $f(x')$ derived in step i). We can also apply a similar argument for the upper bound. $\square$

We compensate for $f \in \widetilde{\mathcal{F}}_\eta$ by adding a buffer of size $r$ and $s$ to the interval derived in Proposition 3.4. If the average lower and upper bound gap is large, then we would have a larger compensation $r, s$. When $\eta = 0$, we have $r = s = 0$. In general, we can bound the buffers $r, s$ in terms of $\eta$.

**Proposition 3.7.** *Under the conditions of Theorem 3.6, we can bound $r_\eta(x)$ and $s_\eta(x)$, as*

$$r_\eta(x) \leq \inf_\delta \delta + (\eta / \Pr(lg_{X \to x}^{(m)} \leq \delta))^{1/p} \quad \text{and} \quad s_\eta(x) \leq \inf_\delta \delta + (\eta / \Pr(ug_{X \to x}^{(m)} \leq \delta))^{1/p}. \tag{13}$$

# 4 Main results

We present our main theoretical results on learning with interval targets. Our analysis proceeds into three steps: first establishing a basic error bound for the realizable setting, then extending it to provide explicit sample complexity guarantees and finally extending it to the agnostic setting. We provide the sample complexity results and their interpretation here and provide the full analysis in Appendix E. The following result is also applicable to $L_p$ loss or a general loss function satisfying Assumption 1 but we state the result for the $l_1$ loss for simplicity.

**Theorem 4.1** (Generalization bound, Realizable Setting). *Let $\mathcal{F}$ be a hypothesis class satisfying i) Realizability and $m$-Lipschitzness, ii) Rademacher complexity decays as $O(1/\sqrt{n})$, iii) support of the distribution $\mathcal{D}_I$ is bounded, iv) loss function is $\ell(y, y') = |y - y'|$. With probability at least $1 - \delta$, for any $f$ that minimize the objective equation 5, for any $\tau > 0$,*

$$\text{err}(f) \leq \underbrace{\mathbb{E}_X[|u^{(m)}_{\mathcal{D} \to X} - l^{(m)}_{\mathcal{D} \to X}|]}_{(a)} + \underbrace{\tau + \left( \frac{D}{\sqrt{n}} + M\sqrt{\frac{\ln(1/\delta)}{n}} \right) \Gamma(\tau)}_{(b)}, \tag{14}$$

*where $D, M$ are constants and $\Gamma(\tau) = \mathbb{E}_{\widetilde{X}}\left[ 1/\min(\Pr_X(lg^{(m)}_{X \to \widetilde{X}} \leq \tau), \Pr_X(ug^{(m)}_{X \to \widetilde{X}} \leq \tau)) \right]$ is decreasing in $\tau$.*

**Interpretation:** Our error bound is divided into two parts.

- (a) The first term represents an **irreducible** error term which depends on the smoothness property of our function class $\mathcal{F}$ and the quality of the given intervals (it does not decrease as $n$ is larger). However, this term can be small. For example, in the case when the ambiguity degree is small, this error term would be zero, ensuring a perfect recovery of the true labels.

- (b) The second and third term capture how well we can learn a hypothesis that belongs to the intervals and these would decay as we have a larger sample size $n$. To see this, assume that we have a fixed value of $\tau$, if one set $n \to \infty$ then the third term would converge to zero. That is, $(b)$ would converge to $\tau$ as $n \to \infty$. Since $\tau$ is arbitrary, we can set $\tau$ to be small so that $(b)$ would decay to zero as $n \to \infty$ and we are left with the first term $(a)$. In addition, the function $\Gamma(\tau)$ depends on the distribution of intervals $\mathcal{D}_I$. In particular, when $\mathcal{D}_I$ has small lower/upper bound gaps, $\Gamma(\tau)$ would also be small which leads to a better generalization bound for any fixed $n$.

**Theorem 4.2** (Generalization Bound, Agnostic Setting). *Under the conditions of Theorem 4.1 apart from realizability, with probability at least $1 - \delta$, for any $f$ that minimize the empirical projection objective, for any $\tau > 0$,*

$$\text{err}(f) \leq \underbrace{\text{OPT}}_{(a)} + \underbrace{\mathbb{E}_X[|u^{(m)}_{\mathcal{D} \to X} - l^{(m)}_{\mathcal{D} \to X}|]}_{(b)} + \underbrace{2\tau + \left( \text{err}_{proj}(f) + \frac{D}{\sqrt{n}} + M\sqrt{\frac{\ln(1/\delta)}{n}} + \text{OPT} \right) \Gamma(\tau)}_{(c)},$$

$$\tag{15}$$

*where $D, M$ are constants and $\Gamma(\tau) = \mathbb{E}_{\widetilde{X}}\left[ 1/\min(\Pr_X(lg^{(m)}_{X \to \widetilde{X}} \leq \tau), \Pr_X(ug^{(m)}_{X \to \widetilde{X}} \leq \tau)) \right]$ is a decreasing function of $\tau$, $\text{err}_{proj}(f)$ is an empirical projection error of $f$, and $\text{OPT}$ is the expected error of the optimal hypothesis in $\mathcal{F}$.*

**Interpretation:** Our error bound for the agnostic setting is divided into three parts.

- (a) The first term represent an error term of the optimal hypothesis in $\mathcal{F}$, given by $\text{OPT}$.

- (b) The second term represent an error term which depends on the smoothness property of our function class $\mathcal{F}$ and the quality of the given intervals similar to the realizability setting.

- (c) The third and the fourth term capture how well we can learn a hypothesis that belongs to the intervals. The key difference between this agnostic setting and the realizability setting is that this term would not decay to zero anymore as $n \to \infty$. In particular, for a fixed $\tau$, we can see that as $n \to \infty$, we would have $\text{err}_{proj}(f) \leq \text{OPT}$ since we are minimizing the empirical projection loss and as a result, this third part would converge to

$$2\tau + 2\,\text{OPT}\cdot\Gamma(\tau). \tag{16}$$

Since this hold for any $\tau$, the optimal $\tau$ would be the one such that $\tau = \text{OPT} \cdot \Gamma(\tau)$ and this value depends on the distribution $\mathcal{D}_I$.

Overall, when $n \to \infty$, the upper bound would converge to

$$\text{OPT} + \mathbb{E}_X[|u_{\mathcal{D} \to X}^{(m)} - l_{\mathcal{D} \to X}^{(m)}|] + 2\tau + 2\,\text{OPT} \cdot \Gamma(\tau). \tag{17}$$

This can be small as long as the OPT is small, the expected lower/upper bound gaps are small and when the noise in the given intervals are small. Overall, our theoretical insight suggests that we can improve our error bound by (i) having a smoother hypothesis class (smaller $m$) which would reduce the interval size $|u_{\mathcal{D} \to X}^{(m)} - l_{\mathcal{D} \to X}^{(m)}|$ in the term (b) (ii) increasing the number of data points $n$ which leads to a smaller bound in the term (c). However, if $m$ is too small (our hypothesis is too smooth), $\mathcal{F}$ may not contain a good hypothesis, causing OPT to be large. Our theoretical results suggest that selecting an appropriate level of smoothness to balance the two terms can lead to improved performance in practice. In practice, we can find the right level of smoothness by treating $m$ as a hyperparameter and tuning it on a validation set.

## 5 Learning from intervals using a minmax objective

In this section, we explore a different learning strategy: we aim to learn a function $f \in \mathcal{F}$ that minimizes the maximum loss with respect to the worst-case $\tilde{y}$ within the interval. We demonstrate that this approach yields a point-wise solution that can be evaluated efficiently. First, we define the worst-case loss as

$$\rho_\ell(f(x), l, u) := \max_{\tilde{y} \in [l,u]} \ell(f(x), \tilde{y}). \tag{18}$$

**Proposition 5.1.** *Let $\ell$ be a loss function that satisfies Assumption 1, then*

$$\rho_\ell(f(x), l, u) = \mathbb{1}[f(x) \leq \frac{l+u}{2}]\ell(f(x), u) + \mathbb{1}[f(x) > \frac{l+u}{2}]\ell(f(x), l). \tag{19}$$

Since $y \in [l, u]$, this objective serves as an upper bound on the true loss: $\rho_\ell(f(x), l, u) \geq \ell(f(x), y)$. Consequently, if we have a hypothesis with a small expected value $\mathbb{E}[\rho_\ell(f(x), l, u)]$, then the error $\text{err}(f)$ will also be small. Based on Proposition 5.1, we define the **Minmax** objective as

$$\min_f \sum_{i=1}^n \mathbb{1}[f(x_i) \leq \frac{l_i+u_i}{2}]\ell(f(x_i), u_i) + \mathbb{1}[f(x_i) > \frac{l_i+u_i}{2}]\ell(f(x_i), l_i). \tag{20}$$

In particular, when $\ell(y, y') = |y - y'|$, we can show that minimizing $\rho$ is equivalent to performing supervised learning using the mid-point of each interval.

**Corollary 5.2.** *Let $\ell(y, y') = |y - y'|$ then $\rho_\ell(f(x), l, u) = |f(x) - \frac{l+u}{2}| + \frac{u-l}{2}$ and the solution of equation 20 is equivalent to*

$$f' = \arg\min_{f \in \mathcal{F}} \sum_{i=1}^n |f(x_i) - \frac{l_i+u_i}{2}|. \tag{21}$$

This corollary establishes a connection between the heuristic of using the midpoint as a target and our approach of minimizing the maximum loss $\rho$. However, we note that $\rho$ does not take the smoothness of the hypothesis class $\mathcal{F}$ into account and may lead to the worst-case labels that are overly conservative and not reflective of the target labels. Therefore, it would be beneficial to incorporate knowledge about certain properties of the true labels. In particular, in the realizable setting, $f^* \in \widetilde{\mathcal{F}}_0$, so we may consider the worst-case labels that can be generated by some $f \in \widetilde{\mathcal{F}}_0$,

$$\min_{f \in \mathcal{F}} \max_{f' \in \widetilde{\mathcal{F}}_0} \mathbb{E}[\ell(f(X), f'(X))]. \tag{22}$$

In the realizable setting, this method also provides an upper bound for $\text{err}(f)$, but it is stronger than $\rho$ because we are comparing against the worst-case $f' \in \widetilde{\mathcal{F}}_0$ rather than any possible $\tilde{y} \in [l, u]$.

**Proposition 5.3.** *In the realizable setting where $f^* \in \widetilde{\mathcal{F}}_0$, for a bounded loss $\ell$, for any $f \in \mathcal{F}$,*

$$\text{err}(f) \leq \max_{f' \in \widetilde{\mathcal{F}}_0} \mathbb{E}[\ell(f(X), f'(X))] \leq \mathbb{E}[\rho_\ell(f(X), L, U)]. \tag{23}$$

We can conclude that when a hypothesis has a small minmax objective, its expected loss would be small as well. Moreover, we demonstrate that restricting the worst-case labels to those that could be generated by some $f \in \widetilde{\mathcal{F}}_0$ can lead to better performance than using all possible worst-case labels. This is due to worst-case labels being highly sensitive to the interval size.

**Proposition 5.4.** *For any constant $c > 0$ and $\ell(y, y') = |y - y'|$, there exists a distribution $\mathcal{D}_I$ and a hypothesis class $\mathcal{F}$ and $f^* \in \mathcal{F}$ such that for $f_1 = \arg\min_{f \in \mathcal{F}} \max_{f' \in \widetilde{\mathcal{F}}_0} \mathbb{E}[\ell(f(X), f'(X)]$ and $f_2 = \arg\min_{f \in \mathcal{F}} \mathbb{E}[\rho_\ell(f(X), L, U)]$, $\mathrm{err}(f_1) = 0$ while $\mathrm{err}(f_2) > c$.*

The proof is in Appendix C.8. An empirical Minmax objective using labels from $\widetilde{\mathcal{F}}_0$ is given by

$$\min_{f \in \mathcal{F}} \max_{f' \in \widetilde{\mathcal{F}}_0} \sum_{i=1}^{n} \ell(f(x_i), f'(x_i)). \tag{24}$$

However, there is no closed-form solution for the inner maximization of objective in 24, making it less efficient to optimize than equation 20. To address this, we propose alternative approaches by approximately learning $f' \in \widetilde{\mathcal{F}}_0$ to solve this objective.

## 5.1 Alternative approaches to solving a minmax objective with constraints

Recall that an empirical Minmax objective using labels from $\widetilde{\mathcal{F}}_0$ is given by equation 24. However, there is no closed-form solution for the inner maximization of objective in 24, making it less efficient to optimize than equation 20. To address this, we propose alternative approaches by approximately learning $f' \in \widetilde{\mathcal{F}}_0$ to solve this objective.

**1) Regularization.** We keep track of two hypothesis $f, f' \in \mathcal{F}$ and introduce a regularization term based on the projection loss to ensure that $f'$ is close $\widetilde{\mathcal{F}}_0$. We call this method **Minmax (reg)**,

$$\min_{f \in \mathcal{F}} \max_{f' \in \mathcal{F}} \sum_{i=1}^{n} \ell(f(x_i), f'(x_i)) - \lambda \sum_{i=1}^{n} \pi(f'(x_i), l_i, u_i). \tag{25}$$

Here the regularization term is always non-positive and depends only on $f'$. We can use a gradient descent ascent [Korpelevich, 1976, Chen and Rockafellar, 1997, Lin et al., 2020] algorithm that updates $f$ and $f'$ with one gradient step at a time to solve this objective.

**2) Pseudo labels.** We could replace a hypothesis class $\widetilde{\mathcal{F}}_0$ with a finite set of hypotheses $\{f_1, f_2, \ldots, f_k\}$ where $f_j \in \widetilde{\mathcal{F}}_\eta$ for some small $\eta$. We can get $f_j$ by minimizing the empirical projection loss. We then relax our objective by learning $f$ that minimizes the maximum loss with respect to $f_j$. We call this method **PL (Max)**,

$$\min_{f \in \mathcal{F}} \max_{j \in \{1, \ldots, k\}} \sum_{i=1}^{n} \ell(f(x_i), f_j(x_i)). \tag{26}$$

Since $f_j$ are fixed, learning $f$ becomes a minimization problem, which is more stable to solve compared to the original minmax problem. Alternatively, to further stabilize the learning objective, we can replace the max over $f_j$ with mean. We refer to this variant as **PL (Mean)**,

$$\min_{f \in \mathcal{F}} \sum_{j=1}^{k} \sum_{i=1}^{n} \ell(f(x_i), f_j(x_i)). \tag{27}$$

## 6 Experiments

We empirically validate our theoretical results with comprehensive experiments on five public datasets from the UCI Machine Learning Repository and 18 additional tabular regression datasets [Grinsztajn et al., 2022], where we vary our proposed interval-generating algorithms to simulate different scenarios and convert regression targets into interval targets (see Appendix G for full details). To control the smoothness of our hypothesis as required by our theoretical results, we utilize Lipschitz MLPs—MLPs augmented with spectral normalization layers [Miyato et al., 2018] that

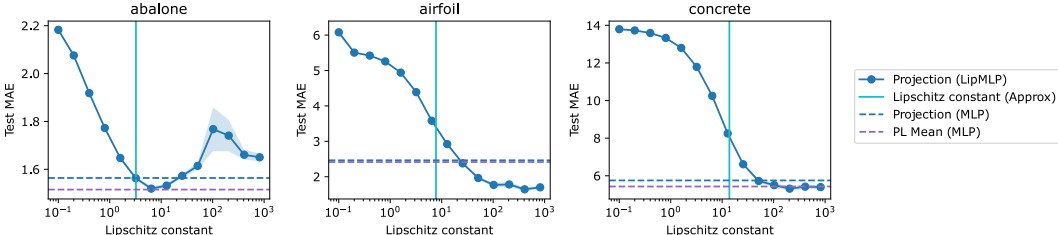

Figure 7: Test MAE of the projection method with Lipschitz MLP using different values of the Lipschitz constant. The vertical line is the Lipschitz constant approximated from the training set. The dashed horizontal lines are the test MAE of PL (Mean) and Projection approach with a standard MLP. Optimal smoothness level leads to a performance gain.

ensure the Lipschitz constant is less than 1, then scaled by a factor of $m$ to control the hypothesis smoothness. We compare standard MLPs against these Lipschitz MLPs, where both model types use projection losses, and we also compare with the minmax loss and our proposed minmax loss variants PL(Mean) and PL(Max). We summarize our findings as follows. In terms of learning methods, the projection objective and our proposed PL methods generally perform best in the uniform interval setting (where interval sizes and locations are uniformly sampled), while naive minmax excels when the target value is known to be near the interval center (consistent with Corollary 5.2). More importantly, we demonstrate that Lipschitz-constrained hypothesis classes indeed achieve smaller reduced intervals, as predicted by Theorem 3.6, with average interval size decreasing as the Lipschitz constant decreases. Our key theoretical insight about the relationship between smoothness and error bounds is supported by experiments showing that the optimal Lipschitz constant balances constraining the hypothesis class while maintaining enough capacity for low error. Finally, on 18 additional tabular regression benchmarks, Lipschitz MLPs significantly outperform standard MLPs on 14 datasets (Table 1), establishing smoothness as a simple yet effective method for enhancing learning with interval targets. Additional results and ablation studies are provided in Appendix L. Our code is available at `https://github.com/bloomberg/interval_targets`.

Table 1: Comparison of the test MAE of LipMLP and MLP results on datasets from the tabular regression benchmark (with interval targets).

| Dataset | LipMLP | MLP | Dataset | LipMLP | MLP |
|---|---|---|---|---|---|
| Ailerons | $\mathbf{3.278 \pm 0.034}$ | $4.323 \pm 0.098$ | Airlines Delay | $\mathbf{38.974 \pm 0.005}$ | $39.077 \pm 0.008$ |
| Allstate Claims | $86.547 \pm 0.001$ | $\mathbf{86.542 \pm 0.002}$ | Analcatdata Supreme | $\mathbf{17.685 \pm 0.041}$ | $17.856 \pm 0.072$ |
| CPU Activity | $\mathbf{10.271 \pm 0.026}$ | $10.560 \pm 0.087$ | Elevators | $\mathbf{59.663 \pm 0.167}$ | $59.926 \pm 0.251$ |
| GPU | $29.817 \pm 0.100$ | $\mathbf{25.123 \pm 0.888}$ | House 16H | $\mathbf{5.728 \pm 0.031}$ | $5.837 \pm 0.025$ |
| House Sales | $\mathbf{76.607 \pm 0.116}$ | $76.716 \pm 0.073$ | Houses | $\mathbf{30.689 \pm 0.152}$ | $31.515 \pm 0.332$ |
| Mercedes | $\mathbf{8.791 \pm 0.187}$ | $11.207 \pm 0.218$ | Miami House | $\mathbf{1.013 \pm 0.028}$ | $1.671 \pm 0.055$ |
| Sulfur | $\mathbf{10.681 \pm 0.082}$ | $14.421 \pm 0.279$ | Superconduct | $\mathbf{0.540 \pm 0.021}$ | $1.459 \pm 0.099$ |
| Topo 21 | $\mathbf{1.305 \pm 0.013}$ | $2.192 \pm 0.177$ | Visualizing Soil | $\mathbf{15.803 \pm 0.311}$ | $17.898 \pm 0.640$ |
| Wine Quality | $\mathbf{28.537 \pm 0.126}$ | $29.537 \pm 0.148$ | YProp 4 | $\mathbf{2.360 \pm 0.050}$ | $3.828 \pm 0.435$ |

## 7    Conclusion

We theoretically investigated the problem of learning from interval targets, analyzing hypotheses that lie within these intervals and those minimizing the worst-case label loss. We derived a novel theoretical bound, providing a crucial insight: understanding how smoothness can lead to benefits such as smaller predictive intervals and a regularized worst-case label. This connection makes our theoretical findings directly applicable in practice. Future directions include more challenging settings such as 'noisy' settings where targets might have small projection loss even outside the interval, and extend these methods to non-i.i.d. settings e.g. time-series.

## Acknowledgments and Disclosure of Funding

Rattana Pukdee is supported by the Bloomberg Data Science Ph.D. Fellowship.

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

# Supplementary Materials:
# Learning with Interval Targets, NeurIPS 2025

## A   Additional related work

**Weak supervision.** Our setting is part of a sub-field of weak supervision where one learns from noisy, limited, or imprecise sources of data rather than a large amount of labeled data. Learning from noisy labels assumes that we only observe a noisy version of the true labels at the training time where the noise follows different noise models (usually random noise) [Natarajan et al., 2013, Li et al., 2017, Song et al., 2022, Angluin and Laird, 1988, Karimi et al., 2020, Awasthi et al., 2017, Chen et al., 2019, Long and Servedio, 2008, Diakonikolas et al., 2019]. Programmatic weak supervision, on the other hand, assumes that we have access to multiple noisy weak labels (but deterministic noise) specified by domain experts, e.g. from logic rules or heuristics methods [Zhang et al., 2022, Zhang et al., Ratner et al., 2016, 2017, Rühling Cachay et al., 2021, Shin et al., 2022, Karamanolakis et al., 2021, Fu et al., 2020, Pukdee et al., 2023b]. Positive-unlabeled learning is another type of weak supervision where the training set only contains positive examples and unlabeled examples [Kiryo et al., 2017, Du Plessis et al., 2014, Bekker and Davis, 2020, Elkan and Noto, 2008, Li and Liu, 2003, Hsieh et al., 2015].

**Learning with side information.** In contrast to the weakly supervised setting, we have access to standard labeled data but also have access to some additional information. This could be unlabeled data which is studied in semi-supervised learning [Zhu, 2005, Chapelle et al., Kingma et al., 2014, Van Engelen and Hoos, 2020, Berthelot et al., 2019, Zhu and Goldberg, 2022, Laine and Aila, 2016, Zhai et al., 2019, Sohn et al., 2020, Yang et al., 2016] or different constraints based on the domain knowledge such as physics rules [Willard et al., 2020, Swischuk et al., 2019, Karniadakis et al., 2021, Wu et al., 2018, Kashinath et al., 2021] or explanations [Ross et al., 2017, Pukdee et al., 2023a, Rieger et al., 2020, Erion et al., 2021] or output constraints [Yang et al., 2020, Brosowsky et al., 2021] which is similar to the interval targets. In some settings, interval targets are the best thing one could have (similar to the weak supervision setting) but in many cases such as in bond pricing, target intervals are readily available in the wild and could also be considered as a side information.

## B   Limitations

Our theoretical results rely on a Lipschitz continuity assumption to characterize the size of the reduced interval. We note that other similar assumptions, such as a modulus of continuity, could also lead to analogous results. Importantly, we do not impose any assumptions on the distribution of the intervals themselves. While this generality can be viewed as a strength, it would be an interesting direction for future work to investigate whether stronger results are possible under additional structural assumptions on the intervals. Our generalization bounds are derived via uniform convergence. This approach is necessary to accommodate general loss functions and hypothesis classes but may be suboptimal compared to specialized analyses—such as those for least squares regression—which do not rely on uniform convergence and can yield sharper rates. For clarity, we assume deterministic labels, although our framework allows for interval targets to be random (see Appendix D). Extending the results to fully random labels is in principle possible, though the notion of correctness—i.e., whether the interval contains the label—becomes less well-defined in such settings. Finally, we assume that the data distribution is nonatomic, which enables us to reason about zero-probability events. This is a standard technical condition that does not limit the applicability of our results to discrete or finite-support distributions.

# C  Additional proofs

## C.1  Proof of Proposition 2.1

*Proof.* First, we assume that $\pi_\ell(f(x), l, u) = 0$. This implies that there exists $\tilde{y} \in [l, u]$ such that $\ell(f(x), \tilde{y}) = 0$. From the assumption on $\ell$ that $\ell(y, y') = 0$ if and only if $y = y'$, we must have $f(x) = \tilde{y} \in [l, u]$ as required. On the other hand, if $f(x) \in [l, u]$, it is clear that $\pi_\ell(f(x), l, u) = \ell(f(x), f(x)) = 0$ since $\ell(y, y') \geq 0$.

Now, assume that we can write $\ell(y, y') = \psi(|y - y'|)$ for some non-decreasing function $\psi$, we have

$$\pi_\ell(f(x), l, u) = \min_{\tilde{y} \in [l,u]} \psi(|f(x) - \tilde{y}|) \tag{28}$$

$$= \psi(\min_{\tilde{y} \in [l,u]} |f(x) - \tilde{y}|) \tag{29}$$

$$= \begin{cases} \psi(l - f(x)) & f(x) < l \\ \psi(0) & l \leq f(x) \leq u \\ \psi(f(x) - u) & f(x) > u \end{cases} \tag{30}$$

$$= 1[f(x) < l]\ell(f(x), l) + 1[f(x) > u]\ell(f(x), u). \tag{31}$$

Here we rely on the assumption that $\psi$ is non-decreasing so the minimum value of $\psi(x)$ happens when $x$ is also at the minimum value. $\square$

## C.2  Proof of Proposition E.3

*Proof.* Since $f_1 \neq f_2$, there exists $x$ such that $f_1(x) \neq f_2(x)$. Without loss of generality, let $f_1(x) < f_2(x)$. Consider a simple one point distribution $\mathcal{D}$ with only one data point $(x, y) = (x, f_2(x) + \epsilon)$ with probability mass 1 and $\mathcal{D}_I$ be another one point distribution with $(x, l, u) = (x, f(x_1) - \epsilon, f(x_2) - \epsilon)$. We can see that $0 = \mathbb{E}_{\mathcal{D}_I}[\pi(f_1(X), L, U)] < \mathbb{E}_{\mathcal{D}_I}[\pi(f_2(X), L, U)] = \epsilon^p$ while $(f(x_2) - f(x_1) + \epsilon)^p = \text{err}(f_1) > \text{err}(f_2) = \epsilon^p$. $\square$

## C.3  Proof of Proposition E.4

*Proof.* From the Proposition 2.1,

$$\pi(f(x), l, u) = 1[f(x) < l]\ell(f(x), l) + 1[f(x) > u]\ell(f(x), u) \tag{32}$$

Recall that $y \in [l, u]$, we consider 3 cases,

1. $f(x) < l$, $\pi(f(x), l, u) = \ell(f(x), l) = \psi(|l - f(x)|) \leq \psi(|y - f(x)|) = \ell(f(x), y)$

2. $f(x) > u$, $\pi(f(x), l, u) = \ell(f(x), u) = \psi(|f(x) - u|) \leq \psi(|f(x) - y|) = \ell(f(x), y)$

3. $l \leq f(x) \leq u$, $\pi(f(x), l, u) = 0 \leq \ell(f(x), y)$

$\square$

## C.4  Proof of Theorem E.5

*Proof.* From the triangle inequality,

$$\ell(f(x), y) = \ell(f(x), f_{\text{opt}}(x)) + \ell(f_{\text{opt}}(x), y) \tag{33}$$

We can take an expectation to have

$$\mathbb{E}[\ell(f(X), Y)] \leq \mathbb{E}[\ell(f(X), f_{\text{opt}}(X)] + \text{OPT}. \tag{34}$$

Since $f_{\text{opt}} \in \widetilde{\mathcal{F}}_{\text{OPT}}$ which from Theorem 3.6, we can bound

$$f_{\text{opt}}(x) \in [l_{\mathcal{D} \to x}^{(m)} - r_{\text{OPT}}(x), l_{\mathcal{D} \to x}^{(m)} + s_{\text{OPT}}(x)]. \tag{35}$$

Similarly, for any $f \in \widetilde{\mathcal{F}}_\eta$, we have

$$f(x) \in [l_{\mathcal{D} \to x}^{(m)} - r_\eta(x), u_{\mathcal{D} \to x}^{(m)} + s_\eta(x)] \tag{36}$$

Finally, we can bound the error between any two intervals with the maximum loss between their boundaries. $\square$

## C.5 Proof of Proposition 5.1

*Proof.* Since we can write $\ell(y, y') = \psi(|y - y'|)$ for some non-decreasing function $\psi$, we have

$$\rho_\ell(f(x), l, u) = \max_{\tilde{y} \in [l,u]} \psi(|f(x) - \tilde{y}|) \tag{37}$$

$$= \psi(\max_{\tilde{y} \in [l,u]} |f(x) - \tilde{y}|) \tag{38}$$

$$= \begin{cases} \psi(u - f(x)) & f(x) < \frac{l+u}{2} \\ \psi(f(x) - l) & f(x) \geq \frac{l+u}{2} \end{cases} \tag{39}$$

$$= 1[f(x) \leq \frac{l+u}{2}]\ell(f(x), u) + 1[f(x) > \frac{l+u}{2}]\ell(f(x), l). \tag{40}$$

Here we rely on the assumption that $\psi$ is non-decreasing so the maximum value of $\psi(x)$ happens when $x$ is also at the maximum value. $\qquad\square$

## C.6 Proof of Corollary 5.2

*Proof.* Since $\ell(y, y') = |y - y'|$, from Proposition 5.1, we have a closed form solution of $\rho$,

$$\rho_\ell(f(x), l, u) = 1[f(x) \leq \frac{l+u}{2}]\ell(f(x), u) + 1[f(x) > \frac{l+u}{2}]\ell(f(x), l) \tag{41}$$

$$= 1[f(x) \leq \frac{l+u}{2}](u - f(x)) + 1[f(x) > \frac{l+u}{2}](f(x) - l) \tag{42}$$

$$= 1[f(x) \leq \frac{l+u}{2}](u - \frac{l+u}{2} + \frac{l+u}{2} - f(x)) + 1[f(x) > \frac{l+u}{2}](f(x) - \frac{l+u}{2} + \frac{l+u}{2} - l) \tag{43}$$

$$= \frac{u-l}{2} + 1[f(x) \leq \frac{l+u}{2}](\frac{l+u}{2} - f(x)) + 1[f(x) > \frac{l+u}{2}](f(x) - \frac{l+u}{2}) \tag{44}$$

$$= |f(x) - \frac{l+u}{2}| + \frac{u-l}{2}. \tag{45}$$

Since $u_i, l_i$ are constants, $\frac{u_i - l_i}{2}$ would have no impact on the optimal solution of equation 20 and therefore, the optimal would also be the same as the one that minimizes $\sum_{i=1}^{n} |f(x_i) - \frac{l_i + u_i}{2}|$. $\qquad\square$

## C.7 Proof of Proposition 5.3

*Proof.* From the realizability assumption, we know that $f^* \in \widetilde{\mathcal{F}}_0$, therefore,

$$\text{err}(f) = \mathbb{E}[\ell(f(X), f^*(X))] \leq \max_{f' \in \widetilde{\mathcal{F}}_0} \mathbb{E}[\ell(f(X), f'(X))]. \tag{46}$$

On the other hand, Let $f'' \in \widetilde{\mathcal{F}}_0$, be a hypothesis that achieves the maximum value of $\mathbb{E}[\ell(f(X), f''(X))]$. Since $f'' \in \widetilde{\mathcal{F}}_0$ we know that

$$\mathbb{E}[\pi_\ell(f''(X), L, U)] = 0. \tag{47}$$

Since the projection loss is always non-negative and is continuous, from Lemma D.1, we can conclude that $\pi_\ell(f''(x), l, u) = 0$ for any $x, l, u$ with positive density function $p(x, l, u) > 0$ which implies $f''(x) \in [l, u]$. Therefore, for any $x$ with $p(x) > 0$,

$$\ell(f(x), f''(x)) \leq \max_{\tilde{y} \in [l,u]} \ell(f(x), \tilde{y}) = \rho_\ell(f(x), l, u). \tag{48}$$

We can take an expectation over $X, L, U$ and have the desired result. $\qquad\square$

## C.8 Proof of Proposition 5.4

*Proof.* Consider when $\mathcal{X} = \{0, 1\}$ and $f^*$ such that $f^*(0) = f^*(1) = 0$. Consider a hypothesis class of constant functions $\mathcal{F} = \{f : \mathcal{X} \to \mathbb{R} \mid f(x) = d, \forall x \in \mathcal{X}\}$. We can see that $f^* \in \mathcal{F}$. Assume that we have a uniform distribution over $\mathcal{X}$ and we also have deterministic interval $[l(x), u(x)]$. Assume

that for $x = 0$, we have an interval $[l(0), u(0)] = [-a, \epsilon]$ for some $a > 0$ and for $x = 1$, we have an interval $[l(1), u(1)] = [-\epsilon, 2\epsilon]$. Since $\mathcal{F}$ is a class of constant hypothesis, for all $x$, we must have $f(x) \in [-a, \epsilon] \cap [-\epsilon, 2\epsilon] = [-\epsilon, \epsilon]$. This implies that

$$\widetilde{\mathcal{F}}_0 = \{f \mid f(x) = c, \forall x \in \mathcal{X}, c \in [-\epsilon, \epsilon]\}. \tag{49}$$

Therefore,

$$f_1 = \arg\min_{f \in \mathcal{F}} \max_{f' \in \widetilde{\mathcal{F}}_0} \mathbb{E}[\ell(f(X), f'(X)] \tag{50}$$

$$= \arg\min_{f \in \mathcal{F}} \max_{f' \in \widetilde{\mathcal{F}}_0} \frac{1}{2}(|f(0) - f'(0)| + |f(1) - f'(1)|) \tag{51}$$

$$= \arg\min_{f \in \mathcal{F}} \max_{c \in [-\epsilon, \epsilon]} |f(0) - c| \tag{52}$$

$$\tag{53}$$

By symmetry, we can see that the optimal $f_1(x) = 0$ which means that $\mathrm{err}(f_1) = 0$. On the other hand, consider $f_2$, from Corollary 5.2, $f_2$ is equivalent to the solution of supervised learning with the midpoint of each interval,

$$f_2 = \arg\min_{f \in \mathcal{F}} \mathbb{E}[\rho_\ell(f(X), L, U)] \tag{54}$$

$$= \arg\min_{f \in \mathcal{F}} \frac{1}{2}[|f(0) - \frac{-a + \epsilon}{2}| + |f(1) - \frac{-\epsilon + 2\epsilon}{2}|]. \tag{55}$$

By symmetry, the optimal $f_2$ should lie in the middle between these two points so that $f_2(x) = -a/2 + \epsilon$. We would have $\mathrm{err}(f_2) = |-a/2 + \epsilon|$ which can be arbitrarily large as $a \to \infty$. $\quad\square$

## D   Probabilistic interval setting

In this section, we consider the probabilistic interval setting which is when, for each $x$, the corresponding interval is drawn from some distribution $\mathcal{D}_I$.

**Assumption 2.** *A distribution $P$ with a probability density function $p(x)$ is a nonatomic distribution when for any $x$ such that $p(x) > 0$ and for any $\epsilon > 0$, there exists a set $S_{x,\epsilon} \subseteq B(x, \epsilon)$ (a ball with radius $\epsilon$) such that $\Pr(S_{x,\epsilon}) > 0$. We assume that the distribution $\mathcal{D}$ and $\mathcal{D}_I$ are nonatomic distributions .*

**Lemma D.1.** *Let $P$ be a nonatomic distribution over $\mathcal{X}$ with a probability density function $p(x)$. For any continuous function $f : \mathcal{X} \to [0, \infty)$, if $\mathbb{E}_P[f(X)] = 0$ then $f(x) = 0$ for all $x$ with $p(x) > 0$.*

*Proof.* We will prove this by contradiction. Assume that there exists $x$ with $p(x) > 0$ such that $f(x) > 0$. By the continuity of $f$, there exists $\delta_1 > 0$ such that for any $x' \in B(x, \delta_1)$ such that $|f(x) - f(x')| \leq f(x)/2$ which implies that $f(x') \geq f(x)/2$. In addition, by the nonatomic assumption, there exists $S_{x,\delta_1} \subseteq B(x, \delta_1)$ such that $\Pr(S_{x,\delta_1}) > 0$. Therefore,

$$\mathbb{E}_P[f(X)] = \int_{w \in \mathcal{X}} f(w)p(w)dw \tag{56}$$

$$\geq \int_{w \in S_{x,\delta_1}} f(w)p(w)dw \tag{57}$$

$$\geq \int_{w \in S_{x,\delta_1}} \frac{f(x)p(w)}{2}dw \tag{58}$$

$$= \frac{f(x)\Pr(S_{x,\delta_1})}{2} > 0. \tag{59}$$

This leads to a contradiction since $\mathbb{E}_P[f(X)] > 0$. $\quad\square$

Similar to the deterministic interval setting, for any $f \in \widetilde{\mathcal{F}}_0$, $f$ has to lie inside the interval as well. One difference would be that in the probabilistic interval setting, we can have multiple intervals for each $x$ and since $f$ has to lie inside all of them, $f$ would also lie inside the intersection of all of them for which we denote as $[\tilde{l}_x, \tilde{u}_x]$ for each $x$.

**Proposition D.2.** *For any $f \in \widetilde{\mathcal{F}}_0$, and a loss function $\ell$ that satisfies Assumption 1, for any $x$ with positive probability density $p(x) > 0$, we have*

$$f(x) \in \bigcap_{p(x,l,u)>0} [l,u] := [\tilde{l}_x, \tilde{u}_x]. \tag{60}$$

*Proof.* Let $f \in \widetilde{\mathcal{F}}_0$ so we have $\mathbb{E}[\pi(f(X), L, U)] = 0$. From Lemma D.1, for any $(x, l, u)$ such that $p(x, l, u) > 0$, we have $\pi(f(x), l, u) = 0$ which implies $f(x) \in [l, u]$ (From Proposition 2.1). Therefore, by taking an intersection over all possible intervals, we would have $f(x) \in \bigcap_{p(x,l,u)>0}[l, u] := [\tilde{l}_x, \tilde{u}_x]$. $\square$

**Proposition D.3.** *Let $\mathcal{F}$ be a class of functions that are $m$-Lipschitz. For any $x, x'$, denote $\tilde{l}^{(m)}_{x' \to x} = \tilde{l}_{x'} - m\|x - x'\|$, $\tilde{u}^{(m)}_{x' \to x} = \tilde{u}_{x'} + m\|x - x'\|$, then for any $f \in \widetilde{\mathcal{F}}_0$ and for any $x$ with positive probability density $p(x) > 0$,*

$$f(x) \in \bigcap_{x'}[\tilde{l}^{(m)}_{x' \to x}, \tilde{u}^{(m)}_{x' \to x}] := [\tilde{l}^{(m)}_{\mathcal{D} \to x}, \tilde{u}^{(m)}_{\mathcal{D} \to x}] \tag{61}$$

*Proof.* Consider $f \in \widetilde{\mathcal{F}}_0$, since $f$ is $m$-Lipschitz, for any $x, x' \in \mathcal{X}$, we have $|f(x) - f(x')| \leq m\|x - x'\|$ which implies

$$f(x') - m\|x - x'\| \leq f(x) \leq f(x') + m\|x - x'\| \tag{62}$$

We illustrate this in Figure 4. Then, from Proposition D.2, for $f \in \widetilde{\mathcal{F}}_0$, we have $\tilde{l}_{x'} \leq f(x') \leq \tilde{u}_{x'}$ which implies

$$\tilde{l}^{(m)}_{x' \to x} = \tilde{l}_{x'} - m\|x - x'\| \leq f(x') - m\|x - x'\| \tag{63}$$

$$\tilde{u}^{(m)}_{x' \to x} = \tilde{u}_{x'} + m\|x - x'\| \geq f(x') - m\|x + x'\|. \tag{64}$$

Substitute back to equation equation 62 and take supremum over $x'$, we have

$$\tilde{l}^{(m)}_{x' \to x} \leq f(x) \leq \tilde{u}^{(m)}_{x' \to x} \tag{65}$$

$$\sup_{x'} \tilde{l}^{(m)}_{x' \to x} \leq f(x) \leq \inf_{x'} \tilde{u}^{(m)}_{x' \to x} \tag{66}$$

$$\tilde{l}^{(m)}_{\mathcal{D} \to x} \leq f(x) \leq \tilde{u}^{(m)}_{\mathcal{D} \to x}. \tag{67}$$

$\square$

Next, we present the probabilistic interval version of Theorem 3.6. Details of the proofs are the same, except that we use $\tilde{l}, \tilde{u}$ instead of $l, u$.

**Theorem D.4.** *Let $\mathcal{F}$ be a class of functions that are $m$-Lipschitz. $\ell : \mathcal{Y} \times \mathcal{Y} \to \mathbb{R}$ is a loss function that satisfies Assumption 1. For any $f \in \widetilde{\mathcal{F}}_\eta$ and for any $x$ with positive probability density $p(x) > 0$,*

$$f(x) \in [\tilde{l}^{(m)}_{\mathcal{D} \to x} - r_\eta(x), \tilde{u}^{(m)}_{\mathcal{D} \to x} + s_\eta(x)] \tag{68}$$

*where $\tilde{l}^{(m)}_{\mathcal{D} \to x}, \tilde{u}^{(m)}_{\mathcal{D} \to x}$ are defined as in Proposition D.3 and*

1. *$r_\eta(x) = r$ such that $\eta = \mathbb{E}[1[g(x, X, r) < L]\ell(g(x, X, r), L)]$ where $g(x, x', r) = \tilde{l}_{x'} - (r - (\tilde{l}^{(m)}_{\mathcal{D} \to x} - \tilde{l}^{(m)}_{x' \to x}))$.*

2. *$s_\eta(x) = s$ such that $\eta = \mathbb{E}[1[h(x, X, s) > U]\ell(h(x, X, s), U)]$ where $h(x, x', s) = \tilde{u}_{x'} + (s - (\tilde{u}^{(m)}_{x' \to x} - \tilde{u}^{(m)}_{\mathcal{D} \to x}))$.*

*Proof.* Now, we will show that if $f \in \widetilde{\mathcal{F}}_\eta$ then we have $f(x) \in [\tilde{l}^{(m)}_{\mathcal{D} \to x} - r_\eta(x), \tilde{u}^{(m)}_{\mathcal{D} \to x} + s_\eta(x)]$ instead. First, we explore what would be a requirement to change the lower bound of $f(x)$ from $\tilde{l}^{(m)}_{\mathcal{D} \to x}$ to $\tilde{l}^{(m)}_{\mathcal{D} \to x} - r$. Again, from Lipschitzness,

$$f(x') - m\|x - x'\| \leq f(x) \tag{69}$$

Taking a supremum here, we have

$$\sup_{x'} f(x') - m\|x - x'\| \leq f(x). \tag{70}$$

Here, we will use $\sup_{x'} f(x') - m\|x - x'\|$ as a new lower bound for $f(x)$. Assume that it is lower than $\tilde{l}_{\mathcal{D}\to x}^{(m)}$, we can write

$$\sup_{x'} f(x') - m\|x - x'\| = \tilde{l}_{\mathcal{D}\to x}^{(m)} - r \tag{71}$$

for some $r > 0$, then it implies that for all $x' \in \mathcal{X}$, we must have

$$f(x') - m\|x - x'\| \leq \tilde{l}_{\mathcal{D}\to x}^{(m)} - r \tag{72}$$

$$(f(x') - \tilde{l}_{x'} + (\tilde{l}_{x'} - m\|x - x'\|)) \leq \tilde{l}_{\mathcal{D}\to x}^{(m)} - r \tag{73}$$

$$f(x') \leq \tilde{l}_{x'} - \tilde{l}_{x'\to x}^{(m)} + \tilde{l}_{\mathcal{D}\to x}^{(m)} - r \tag{74}$$

$$f(x') \leq \tilde{l}_{x'} - (r - (\tilde{l}_{\mathcal{D}\to x}^{(m)} - \tilde{l}_{x'\to x}^{(m)})) \tag{75}$$

That is, if one can change the lower bound of $f(x)$ from $\tilde{l}_{\mathcal{D}\to x}^{(m)}$ to $\tilde{l}_{\mathcal{D}\to x}^{(m)} - r$ then for all $x'$, $f(x')$ has to take value lower than $\tilde{l}_{x'}$ by at least $r - (\tilde{l}_{\mathcal{D}\to x}^{(m)} - \tilde{l}_{x'\to x}^{(m)})$ whenever this term is positive. However, $f \in \tilde{\mathcal{F}}_\eta$ so that $f(x')$ can't be too far away from $\tilde{l}_{x'}$ since $\mathbb{E}[\pi_\ell(f(X), L, U)] \leq \eta$. From Proposition 2.1, if one can write $\ell(y, y') = \psi(|y - y'|)$ for some non-decreasing function $\psi$ then we have

$$\pi_\ell(f(x), l, u) = 1[f(x) < l]\ell(f(x), l) + 1[f(x) > u]\ell(f(x), u). \tag{76}$$

Therefore,

$$\eta \geq \mathbb{E}[\pi_\ell(f(X), L, U)] \geq \mathbb{E}[1[f(X) < L]\ell(f(X), L)]. \tag{77}$$

Let $g(x, x', r) = \tilde{l}_{x'} - (r - (\tilde{l}_{\mathcal{D}\to x}^{(m)} - \tilde{l}_{x'\to x}^{(m)}))$ be the upper bound of $f(x')$ for any $x'$ as we derived in the equation equation 75. Since $1[a < L]\ell(a, L)]$ is a decreasing function over $a$, equation equation 77 implies

$$\eta \geq \mathbb{E}[1[f(X) < L]\ell(f(X), L)] \geq \mathbb{E}[1[g(x, X, r) < L]\ell(g(x, X, r), L)] \tag{78}$$

We can also see that $g(x, x', r)$ is a decreasing function of $r$ which means $\mathbb{E}[1[g(x, X, r) < L]\ell(g(x, X, r), L)]$ is an increasing function of $r$. The largest possible value of $r$ would then be the $r$ such that the inequality holds,

$$\eta = \mathbb{E}[1[g(x, X, r) < L]\ell(g(x, X, r), L)]. \tag{79}$$

which we denoted this as $r_\eta(x)$. Similarly, we can show that if the largest possible value of $s$ such that we can change the upper bound of $f(x)$ from $\tilde{u}_{\mathcal{D}\to x}^{(m)}$ to $\tilde{u}_{\mathcal{D}\to x}^{(m)} + s$ is given by

$$\eta = \mathbb{E}[1[h(x, X, s) > U]\ell(h(x, X, s), U)] \tag{80}$$

where $h(x, x', s) = \tilde{u}_{x'} + (s - (\tilde{u}_{x'\to x}^{(m)} - \tilde{u}_{\mathcal{D}\to x}^{(m)}))$. □

**Theorem D.5.** *Under the conditions of Theorem D.4, if further assume that for each $x$, the lower and upper bound of $y$ is given by deterministic function $[l(x), u(x)]$ and $\ell$ is an $\ell_p$ loss $\ell(y, y') = |y - y'|^p$ and denote the lower bound gap and upper bound gap of $f(x)$ induced by $x'$ as $lg_{x'\to x}^{(m)} = \tilde{l}_{\mathcal{D}\to x}^{(m)} - \tilde{l}_{x'\to x}^{(m)}$ and $ug_{x'\to x}^{(m)} = \tilde{u}_{x'\to x}^{(m)} - \tilde{u}_{\mathcal{D}\to x}^{(m)}$ then we have*

$$r_\eta(x) = r \quad s.t. \quad \mathbb{E}[(r - lg_{X\to x}^{(m)})_+^p] = \eta \tag{81}$$

$$s_\eta(x) = s \quad s.t. \quad \mathbb{E}[(s - ug_{X\to x}^{(m)})_+^p] = \eta \tag{82}$$

*where we denote $c_+ = \max(0, c)$. Further, we can bound $r_\eta(x)$ and $s_\eta(x)$,*

$$r_\eta(x) \leq \inf_\delta \delta + \left(\frac{\eta}{\Pr(lg_{X\to x}^{(m)} \leq \delta)}\right)^{1/p} \tag{83}$$

$$s_\eta(x) \leq \inf_\delta \delta + \left(\frac{\eta}{\Pr(ug_{X\to x}^{(m)} \leq \delta)}\right)^{1/p}. \tag{84}$$

*Proof.* Since $[l, u]$ is deterministic for each $x$, we have $\tilde{l}_x = l(x)$. By the property of squared loss,

$$\mathbb{E}[1[g(x, X, r) < L]\ell(g(x, X, r), L)] = \mathbb{E}[(L - g(x, X, r))_+^p] \tag{85}$$

$$= \mathbb{E}[(l(X) - g(x, X, r))_+^p] \tag{86}$$

$$= \mathbb{E}[(l(X) - (\tilde{l}_X - (r - (\tilde{l}_{\mathcal{D} \to x}^{(m)} - \tilde{l}_{X \to x}^{(m)}))))_+^p] \tag{87}$$

$$= \mathbb{E}[(r - lg_{X \to x}^{(m)})_+^p] \tag{88}$$

as required. We can use a similar argument for $s_\eta(x)$. Next, we can see that for any valid value of $r$,

$$\eta \geq \mathbb{E}[(r - lg_{X \to x}^{(m)})_+^p] \geq \mathbb{E}[(r - \delta)_+^p 1[lg_{X \to x}^{(m)} \leq \delta]] = (r - \delta)_+^p \Pr(lg_{X \to x}^{(m)} \leq \delta). \tag{89}$$

By rearranging, $r \leq \delta + (\frac{\eta}{\Pr(lg_{X \to x}^{(m)} \leq \delta)})^{1/p}$. Taking the infimum over $\delta$, we have the desired inequality. Again, we can apply the same idea for $s_\eta(x)$. $\square$

# E   Sample complexity bounds

## E.1   Error bound in the realizable setting

We begin with a foundational result that characterizes the error of any hypothesis in $\widetilde{\mathcal{F}}_\eta$ based on the reduced intervals established in the previous section.

**Theorem E.1** (Error bound, Realizable setting). *Let $\mathcal{F}$ be a class of functions that are $m$-Lipschitz, assume that $f^* \in \widetilde{\mathcal{F}}_0$, then for any $f \in \widetilde{\mathcal{F}}_\eta$,*

$$\mathrm{err}(f) \leq \mathbb{E}[d(\ell, I_0(X), I_\eta(X))]. \tag{90}$$

*when $I_\eta(x) := [l_{\mathcal{D} \to x}^{(m)} - r_\eta(x), u_{\mathcal{D} \to x}^{(m)} + s_\eta(x)]$ represents the reduced interval from Theorem 3.6 and $d(\ell, I_1, I_2) = \max(\ell(l_1, u_2), \ell(u_1, l_2))$ when $I_1 = [l_1, u_1], I_2 = [l_2, u_2]$.*

We remark that this bound can be tight for certain hypothesis classes. For example, consider the case where $\mathcal{F}$ consists of constant hypotheses and let $n \to \infty$. In this scenario, we have $r_\eta(x) \to r_0(x) = 0$ and $I_\eta(x) \to I_0(x)$. For each $x$, the error bound is given by

$$d(\ell, I_0(x), I_0(x)) = \ell(l_{\mathcal{D} \to x}^{(m)}, u_{\mathcal{D} \to x}^{(m)}) = \ell(\sup_{x'} l_{x'}, \inf_{x'} u_{x'}), \tag{91}$$

representing the loss between the boundaries of the intersected intervals. It is tight since the inequality holds when $f^*$ and $f$ each take values at the respective boundaries of the intersected interval.

## E.2   Main sample complexity result

Building on Theorem E.1, we now present our main result, which provides explicit sample complexity guarantees for learning with interval targets for any hypothesis classes whose the Rademacher complexity decay as $O(1/\sqrt{n})$. This includes a class of linear models or a class of two-layer neural networks with a bounded weight [Ma, 2022]. To simplify the Theorem, we will only present the statement and the proof for the case of $L_1$ loss. However, an extension for a general $L_p$ loss is straightforward where we can replace the triangle inequality with the Minkowski's inequality.

**Theorem E.2** (Generalization bound, Realizable Setting). *Let $\mathcal{F}$ be a hypothesis class satisfying i) the conditions of Theorem E.1 (realizability and $m$-Lipschitzness), ii) Rademacher complexity decays as $O(1/\sqrt{n})$, iii) support of the distribution $\mathcal{D}_I$ is bounded, iv) loss function is $\ell(y, y') = |y - y'|$. With probability at least $1 - \delta$, for any $f$ that minimize the objective equation 5, for any $\tau > 0$,*

$$\mathrm{err}(f) \leq \underbrace{\mathbb{E}_X[|u_{\mathcal{D} \to X}^{(m)} - l_{\mathcal{D} \to X}^{(m)}|]}_{(a)} + \tau + \underbrace{\left( \frac{D}{\sqrt{n}} + M\sqrt{\frac{\ln(1/\delta)}{n}} \right) \Gamma(\tau)}_{(b)}, \tag{92}$$

*where $D, M$ are constants and $\Gamma(\tau) = \mathbb{E}_{\widetilde{X}}\left[ 1/\min(\Pr_X(lg_{X \to \widetilde{X}}^{(m)} \leq \tau), \Pr_X(ug_{X \to \widetilde{X}}^{(m)} \leq \tau)) \right]$ is decreasing in $\tau$.*

*Proof.* **Step 1: Derive the bound in term of $\eta$.** Recall that from Theorem E.1, we have

$$\text{err}(f) \leq \mathbb{E}[d(\ell, I_0(X), I_\eta(X))]. \tag{93}$$

when $I_\eta(x) = [l_{\mathcal{D} \to x}^{(m)} - r_\eta(x), u_{\mathcal{D} \to x}^{(m)} + s_\eta(x)]$. Since we have an $\ell_1$ loss, we have

$$d(\ell, I_0(x), I_\eta(x)) = |u_{\mathcal{D} \to x}^{(m)} - l_{\mathcal{D} \to x}^{(m)} + \max(r_\eta(x), s_\eta(x))|. \tag{94}$$

Substitute this back in, we have an error bound

$$\text{err}(f) \leq \mathbb{E}[|u_{\mathcal{D} \to X}^{(m)} - l_{\mathcal{D} \to X}^{(m)} + \max(r_\eta(X), s_\eta(X))|] \tag{95}$$

$$\leq \mathbb{E}[|u_{\mathcal{D} \to X}^{(m)} - l_{\mathcal{D} \to X}^{(m)}|] + \mathbb{E}[|\max(r_\eta(X), s_\eta(X))|] \quad \text{(triangle inequality).} \tag{96}$$

Now, our goal is to bound the term $\mathbb{E}[|\max(r_\eta(X), s_\eta(X))|]$. From Proposition 3.7, we know that

$$r_\eta(x) \leq \inf_\tau \tau + (\eta / \Pr(lg_{X \to x}^{(m)} \leq \tau)) \quad \text{and} \quad s_\eta(x) \leq \inf_\tau \tau + (\eta / \Pr(ug_{X \to x}^{(m)} \leq \tau)). \tag{97}$$

We place $\delta$ with $\tau$ in the original statement because we will use $\delta$ as something else, later. This implies that

$$\max(r_\eta(x), s_\eta(x)) \leq \inf_\tau \tau + \left( \frac{\eta}{\min(\Pr(lg_{X \to x}^{(m)} \leq \tau), \Pr(ug_{X \to x}^{(m)} \leq \tau))} \right). \tag{98}$$

We define $\Lambda(\mathcal{D}, \tau) = \min(\Pr(lg_{X \to x}^{(m)} \leq \tau), \Pr(ug_{X \to x}^{(m)} \leq \tau))^{-1}$ so that

$$\max(r_\eta(x), s_\eta(x)) \leq \inf_\tau \tau + \eta \Lambda(\mathcal{D}, \tau). \tag{99}$$

We can see that when $\Lambda(\mathcal{D}, \tau) \geq 0$ and $\Lambda(\mathcal{D}, \tau)$ is a decreasing function in $\tau$. Substitue this back to the equation 96, for any $\tau > 0$, we would have

$$\text{err}(f) \leq \mathbb{E}[|u_{\mathcal{D} \to X}^{(m)} - l_{\mathcal{D} \to X}^{(m)}|] + \mathbb{E}[|\tau + \eta \Lambda(\mathcal{D}, \tau)|] \tag{100}$$

$$\leq \mathbb{E}[|u_{\mathcal{D} \to X}^{(m)} - l_{\mathcal{D} \to X}^{(m)}|] + \tau + \eta \mathbb{E}[\Lambda(\mathcal{D}, \tau)] \tag{101}$$

$$= \mathbb{E}[|u_{\mathcal{D} \to X}^{(m)} - l_{\mathcal{D} \to X}^{(m)}|] + \tau + \eta \Gamma(\mathcal{D}, \tau) \tag{102}$$

where we define $\Gamma(\mathcal{D}, \tau) = \mathbb{E}[\Lambda(\mathcal{D}, \tau)]$. We can see that every term in the equation above is independent of $\eta$, apart from the term $\eta$ itself. This provide a more explicit error bound in term of $\eta$. Now, we will bound $\eta$ in terms of the number of sample $n$.

**Step 2: Bounding $\eta$ in terms of the number of sample.** Recall the result from equation 6, with probability at least $1 - \delta$ over the draws $(x_i, l_i, u_i) \sim \mathcal{D}_I$, for all $f \in \mathcal{F}$,

$$\mathbb{E}[\pi_\ell(f(X), L, U)] \leq \frac{1}{n} \sum_{i=1}^n \pi_\ell(f(x_i), l_i, u_i) + 2R_n(\Pi(\mathcal{F})) + M \sqrt{\frac{\ln(1/\delta)}{n}}. \tag{103}$$

Here, $R_n(\Pi(\mathcal{F}))$ is the Rademacher complexity of the function class $\Pi(\mathcal{F}) := \{\pi_\ell(f(x), l, u) \mapsto \mathbb{R} \mid f \in \mathcal{F}\}$ and we assume that the $\pi_\ell$ is uniformly bounded by $M$. We recall that we learn $\hat{f}$ by minimizing the empirical projection loss

$$\hat{f} = \arg\min_{f \in \mathcal{F}} \sum_{i=1}^n \pi_\ell(f(x_i), l_i, u_i). \tag{104}$$

Under the realizable setting, this objective would be zero since $f* \in \mathcal{F}$ which implies that $f*$ has zero empirical projection $\sum_{i=1}^n \pi_\ell(f^*(x_i), l_i, u_i) = 0$ but $\hat{f}$ also minimize the empirical projection loss so $\hat{f}$ must also have a zero empirical projection loss. We write $\eta(f)$ to refer to the $\eta$ value of $f$. Formally, defined as

$$\eta(f) = \mathbb{E}[\pi_\ell(f(X), L, U)]. \tag{105}$$

Substituting $\hat{f}$ to the bound above, we have

$$\eta(\hat{f}) \leq 2R_n(\Pi(\mathcal{F})) + M \sqrt{\frac{\ln(1/\delta)}{n}}. \tag{106}$$

The next step is to bound the Rademacher complexity $R_n(\Pi(\mathcal{F}))$ in terms of $R_n(\mathcal{F})$. We will do this by first showing that $\phi_i(f(x)) = \pi_\ell(f(x), l_i, u_i)$ is a Lipschitz continuous function and then reduce $R_n(\Pi(\mathcal{F}))$ to $R_n(\mathcal{F})$ with a variant of Talagrand's Lemma [Meir and Zhang, 2003]. From our assumption that the support of $\mathcal{D}_I$ is a bounded set, and our hypothesis class is a class of two-layer neural network with bounded weight, there exists a constant $C$ for which, we have $|f(x)| \leq C$ almost surely. Here, we will show this property for $L_p$ loss, recall that

$$\phi_i(f(x)) = \pi_\ell(f(x), l_i, u_i) \tag{107}$$
$$= (l_i - f(x))^p 1[f(x) < l_i] + (f(x) - u_i)^p 1[f(x) > u]. \tag{108}$$

Differentiate with respect to $f(x)$, we have

$$|\nabla_{f(x)} \phi_i(f(x))| = p|(l_i - f(x))^{p-1} 1[f(x) < l_i] + (f(x) - u_i)^{p-1} 1[f(x) > u]| \tag{109}$$
$$\leq 2p(2C)^{p-1}. \tag{110}$$

Since this gradient is bounded for any $f(x)$, we can conclude that $\phi_i(f(x))$ is $B$-Lipschitz for some constant $B$. Now, we unpack the definition of the Rademacher complexity,

$$R_n(\Pi(\mathcal{F})) = \mathbb{E}_{(x_i, l_i, u_i) \sim \mathcal{D}_I}[\mathbb{E}_{\sigma_i \sim \{-1,1\}}[\sup_{f \in \mathcal{F}} \frac{1}{n} \sum_{i=1}^n \pi_\ell(f(x_i), l_i, u_i) \sigma_i]] \tag{111}$$

$$= \mathbb{E}_{(x_i, l_i, u_i) \sim \mathcal{D}_I}[\mathbb{E}_{\sigma_i \sim \{-1,1\}}[\sup_{f \in \mathcal{F}} \frac{1}{n} \sum_{i=1}^n \phi_i(f(x_i)) \sigma_i]]. \tag{112}$$

We recall the following result from Meir and Zhang [2003] that when $\phi_1, \phi_2, \ldots \phi_n$ be functions where $\phi_i : \mathbb{R} \to \mathbb{R}$ are $\phi_i$ are $L_i$-Lipschitz, then

$$\mathbb{E}_{\sigma_i \sim \{-1,1\}}[\sup_{f \in \mathcal{F}} \frac{1}{n} \sum_{i=1}^n \phi_i(f(x_i)) \sigma_i] \leq \mathbb{E}_{\sigma_i \sim \{-1,1\}}[\sup_{f \in \mathcal{F}} \frac{1}{n} \sum_{i=1}^n L_i f(x_i) \sigma_i]. \tag{113}$$

Applying this result with the fact that $\phi_i$ is $B$-Lipschitz for all $i = 1, \ldots, n$, we can conclude that

$$R_n(\Pi(\mathcal{F})) = \mathbb{E}_{(x_i, l_i, u_i) \sim \mathcal{D}_I}[\mathbb{E}_{\sigma_i \sim \{-1,1\}}[\sup_{f \in \mathcal{F}} \frac{1}{n} \sum_{i=1}^n \phi_i(f(x_i)) \sigma_i]] \tag{114}$$

$$\leq \mathbb{E}_{(x_i, l_i, u_i) \sim \mathcal{D}_I}[\mathbb{E}_{\sigma_i \sim \{-1,1\}}[\sup_{f \in \mathcal{F}} \frac{1}{n} \sum_{i=1}^n B f(x_i) \sigma_i]] \tag{115}$$

$$= B R_n(\mathcal{F}). \tag{116}$$

We successfully reduce the Rademacher complexity of $\Pi(\mathcal{F})$ to $\mathcal{F}$. Since we assume that the Rademacher complexity of $\mathcal{F}$ decays as $O(1/\sqrt{n})$, there exists a constant $D$ such that

$$R_n(\Pi(\mathcal{F})) \leq \frac{D}{\sqrt{n}} \tag{117}$$

and

$$\eta(\hat{f}) \leq \frac{D}{\sqrt{n}} + M \sqrt{\frac{\ln(1/\delta)}{n}} \tag{118}$$

for some constant $D, M$. Substitute this back to the result from step 1 concludes our proof. In the general setting with $L_p$ loss where $\ell(y, y') = |y - y'|^p$, we would have the following bound,

$$\text{err}(f) \leq \left( \mathbb{E}_X[|u_{\mathcal{D} \to X}^{(m)} - l_{\mathcal{D} \to X}^{(m)}|^p]^{1/p} + \tau + \left( \frac{D}{\sqrt{n}} + M \sqrt{\frac{\ln(1/\delta)}{n}} \right)^{1/p} \Gamma(\tau)^{1/p} \right)^p \tag{119}$$

$\square$

### E.3 Agnostic setting

Now, we study the agnostic setting, where we do not assume the existence of such $f^*$ in $\mathcal{F}$. Instead, we focus on comparing with $f_{\text{opt}} = \arg\min_{f \in \mathcal{F}} \text{err}(f)$, the hypothesis in $\mathcal{F}$ with the smallest expected error. First, we show that, in contrast to the realizable setting, simply minimizing the projection loss may not converge to $f_{\text{opt}}$. This is because a smaller projection loss $\pi$ does not imply a smaller standard loss $\ell$.

**Proposition E.3.** *Let $\ell$ be an $\ell_p$ loss, for any hypothesis $f_1, f_2$, there exists a distribution $\mathcal{D}_I$ and $\mathcal{D}$ such that $\mathbb{E}_{\mathcal{D}_I}[\pi_\ell(f_1(X), L, U)] < \mathbb{E}_{\mathcal{D}_I}[\pi_\ell(f_2(X), L, U)]$ but $\text{err}(f_1) > \text{err}(f_2)$.*

While minimizing the projection loss, we might overlook a hypothesis that has a smaller standard loss but a higher projection loss. However, we remark that the projection loss is still useful since it is a lower bound of the standard loss.

**Proposition E.4.** *Let $\ell : \mathcal{Y} \times \mathcal{Y} \to \mathbb{R}$ be a loss function that satisfies Assumption 1, then for any $f$,*

$$\mathbb{E}[\pi_\ell(f(X), L, U)] \leq \text{err}(f). \tag{120}$$

Consequently, if we let $\text{OPT} = \text{err}(f_{\text{opt}})$, we must have $f_{\text{opt}} \in \widetilde{\mathcal{F}}_{\text{OPT}}$ since the projection loss is upper bound by the standard loss. This means we can apply Theorem 3.6 for $f_{\text{opt}}$ and consequently achieve an error bound similar to what we obtained in the realizable setting.

**Theorem E.5** (Error bound, Agnostic setting). *Let $\mathcal{F}$ be a class of functions that are $m$-Lipschitz, and suppose $\ell$ satisfies Assumption 1 and the triangle inequality, then for any $f \in \widetilde{\mathcal{F}}_\eta$, we have*

$$\text{err}(f) \leq \text{OPT} + \mathbb{E}[d(\ell, I_\eta(X), I_{\text{OPT}}(X))]. \tag{121}$$

While it's not ideal to minimize the projection loss in the agnostic setting since we may not converge to $f_{\text{opt}}$, our bound suggests that the expected error of $f$ would not be much larger than that of $f_{\text{opt}}$. This error bound becomes smaller when the intervals $I_\eta(x), I_{\text{OPT}}(x)$ are small. Overall, our theoretical insight suggests that we can improve our error bound by (i) having a smoother hypothesis class (smaller $m$) (ii) increasing the number of data points $n$ (which leads to smaller $\eta$), since both results in smaller intervals $I_\eta(x)$. However, if $m$ is too small, $\mathcal{F}$ may not contain a good hypothesis, causing OPT to be large. Next, we provide a sample complexity bound for the agnostic setting.

**Theorem E.6** (Generalization Bound, Agnostic Setting). *Under the conditions of Theorem 4.1 apart from realizability, with probability at least $1 - \delta$, for any $f$ that minimize the empirical projection objective, for any $\tau > 0$,*

$$\text{err}(f) \leq \underbrace{\text{OPT}}_{(a)} + \underbrace{\mathbb{E}_X[|u_{\mathcal{D} \to X}^{(m)} - l_{\mathcal{D} \to X}^{(m)}|]}_{(b)} + 2\tau + \underbrace{\left(\text{err}_{proj}(f) + \frac{D}{\sqrt{n}} + M\sqrt{\frac{\ln(1/\delta)}{n}} + \text{OPT}\right)\Gamma(\tau)}_{(c)},$$

$$\tag{122}$$

*where $D, M$ are constants and $\Gamma(\tau) = \mathbb{E}_{\widetilde{X}}\left[1/\min(\Pr_X(lg_{X \to \widetilde{X}}^{(m)} \leq \tau), \Pr_X(ug_{X \to \widetilde{X}}^{(m)} \leq \tau))\right]$ is a decreasing function of $\tau$, $\text{err}_{proj}(f)$ is an empirical projection error of $f$, and $\text{OPT}$ is the expected error of the optimal hypothesis in $\mathcal{F}$.*

*Proof.* The proof idea is similar to the realizable setting. Recall that we have an error bound

$$\text{err}(f) \leq \text{OPT} + \mathbb{E}[d(\ell, I_\eta(X), I_{\text{OPT}}(X))] \tag{123}$$

where $I_\eta(x) = [l_{\mathcal{D} \to x}^{(m)} - r_\eta(x), u_{\mathcal{D} \to x}^{(m)} + s_\eta(x)]$. We can write

$$d(l, I_\eta(X), I_{\text{OPT}}(X)) \leq |u_{\mathcal{D} \to x}^{(m)} - l_{\mathcal{D} \to x}^{(m)} + \max(r_\eta(x) + s_{\text{OPT}}(x), r_{\text{OPT}}(x) + s_\eta(x))|. \tag{124}$$

With a triangle inequality, substitute this back to the error bound, we have

$$\text{err}(f) \leq \text{OPT} + \mathbb{E}[|u_{\mathcal{D} \to X}^{(m)} - l_{\mathcal{D} \to X}^{(m)}|] + \mathbb{E}[\max(r_\eta(x) + s_{\text{OPT}}(x), r_{\text{OPT}}(x) + s_\eta(x))]. \tag{125}$$

We can see that the first two terms are term a) and b) in the Theorem 4.2. Therefore, we are left with bounding the final term. From Proposition 3.7, we know that for any $\tau > 0$,

$$r_\eta(x) \leq \inf_\tau \tau + (\eta/\Pr(lg_{X \to x}^{(m)} \leq \tau)) \quad \text{and} \quad s_\eta(x) \leq \inf_\tau \tau + (\eta/\Pr(ug_{X \to x}^{(m)} \leq \tau)). \tag{126}$$

This implies that

$$r_\eta(x) + s_{\text{OPT}}(x) \leq 2\tau + (\eta/\Pr(lg_{X\to x}^{(m)} \leq \tau)) + (\text{OPT}/\Pr(ug_{X\to x}^{(m)} \leq \tau)) \tag{127}$$

$$=\leq 2\tau + (\eta + \text{OPT})(\max(1/\Pr(lg_{X\to x}^{(m)} \leq \tau)), 1/\Pr(ug_{X\to x}^{(m)} \leq \tau))) \tag{128}$$

$$=\leq 2\tau + (\eta + \text{OPT})(1/\min(\Pr(lg_{X\to x}^{(m)} \leq \tau), \Pr(ug_{X\to x}^{(m)} \leq \tau)). \tag{129}$$

We have the same upper bound for $r_{\text{OPT}}(x) + s_\eta(x))$. Taking an expectation, we have

$$\mathbb{E}[\max(r_\eta(x) + s_{\text{OPT}}(x), r_{\text{OPT}}(x) + s_\eta(x))] \leq 2\tau + (\eta + \text{OPT})\Gamma(\tau) \tag{130}$$

when $\Gamma(\tau) = \mathbb{E}_{\widetilde{X}}\left[1/\min(\Pr_X(lg_{X\to\widetilde{X}}^{(m)} \leq \tau), \Pr_X(ug_{X\to\widetilde{X}}^{(m)} \leq \tau))\right]$. The final step is to bound $\eta$ in terms of the empirical loss, following the uniform convergence argument from the realizable setting, with probability at least $1 - \delta$,

$$\eta \leq \widehat{\text{err}}(f) + \frac{D}{\sqrt{n}} + M\sqrt{\frac{\ln(1/\delta)}{n}}. \tag{131}$$

This concludes our proof for the agnostic setting. $\qquad\square$

# F   Relaxation of Ambiguity Degree for a regression setting

As noted in the related work section, the ambiguity degree is defined in the context of classification and it might not be suitable for regression tasks. This is due to the nature of the loss function, In classification, a hypothesis is either correct or incorrect, and a small ambiguity degree ensures that we can recover the true label. However, in regression, we are often satisfied with predictions that are sufficiently close to the target—for example, within an error tolerance of $\epsilon$. This implies that we do not need to recover the exact true label, but a ball with a small radius around the true label might be sufficient.

In this section, we explore a relaxation of the original ambiguity degree to the regression setting. Motivated by the concept of a tolerable area around the true label $y$, we define an ambiguity radius

**Definition F.1** (Ambiguity Radius). *For distributions $\mathcal{D}, \mathcal{D}_I$ with a probability density function $p$, an ambiguity radius is defined as*

$$\text{AmbiguityRadius}(\mathcal{D}, \mathcal{D}_I) := \min_{r \geq 0} r \quad s.t. \quad \Pr_{X,Y\sim\mathcal{D}}\Big(\bigcap_{p(X,l,u)>0} [l, u] \subseteq B(Y, r)\Big) = 1 \tag{132}$$

*when $B(y, r) = \{y' \mid |y - y'| \leq r\}$ is a ball of radius $r$ around $y$.*

The interpretation of this is that it is the smallest radius $r$ for which we are guaranteed the intersection of all interval for a given $x$ must lie within a radius of $r$ from the true label $y$. As a direct consequence, we know that whenever the ambiguity degree is small the ambiguity radius must be zero since the intersection of all interval for a given $x$ is just the true label $\{y\}$.

In fact, our analysis have captured the essence of this interval intersection for each $x$. We recall that for any $f \in \widetilde{\mathcal{F}}_0$ and for each $x$ with $p(x) > 0$,

$$f(x) \in I_0(x) = [l_{\mathcal{D}\to x}^{(m)}, u_{\mathcal{D}\to x}^{(m)}] \subseteq B(y, r^*), \tag{133}$$

when $r^*$ is the ambiguity radius. This follows directly from the definition of the ambiguity radius. As a result, we know that each interval $I_0(x)$ would have a size at most $2r^*$. The same technique as in the Section 4 would imply that the expected error of any $f \in \widetilde{\mathcal{F}}_0$ would be at most $2r^*$ in the realizable setting (with $L_1$ loss).

Finally, we want to remark that our analysis not only is applicable to this extension of the ambiguity degree to the ambiguity radius, we further use the smooth property of $\mathcal{F}$ and $I_0(x)$ might even be a proper subset of the ball $B(y, r^*)$, giving a result stronger than one based solely on the ambiguity radius.

|  | Projection (equation 5) | Minmax (equation 20) | Minmax (reg) (equation 25) | PL (max) (equation 26) | PL (mean) (equation 27) |
|---|---|---|---|---|---|
| Abalone | $1.56_{0.01}$ | $1.65_{0.02}$ | $1.54_{0.01}$ | $\mathbf{1.52_{0.01}}$ | $\mathbf{1.52_{0.01}}$ |
| Airfoil | $\mathbf{2.46_{0.08}}$ | $2.65_{0.07}$ | $3.41_{0.04}$ | $3.31_{0.04}$ | $\mathbf{2.42_{0.07}}$ |
| Concrete | $5.75_{0.13}$ | $7.34_{0.2}$ | $6.23_{0.16}$ | $5.86_{0.48}$ | $\mathbf{5.43_{0.12}}$ |
| Housing | $\mathbf{5.17_{0.13}}$ | $6.88_{0.31}$ | $5.42_{0.15}$ | $\mathbf{5.07_{0.09}}$ | $\mathbf{5.05_{0.09}}$ |
| Power-plant | $3.4_{0.03}$ | $3.47_{0.02}$ | $3.48_{0.03}$ | $\mathbf{3.33_{0.01}}$ | $\mathbf{3.33_{0.01}}$ |
| Average (rank) | 2.8 | 4.4 | 4.2 | 2.2 | 1 |

Table 2: Test Mean Absolute Error (MAE) and the standard error (over 10 random seeds) for the uniform interval setting. PL (mean) is the best-performing method in this setting.

# G   Experiments

## G.1   Computational efficiency

The computational cost of our projection objective matches standard regression loss, as we only evaluate boundaries of the given interval (Proposition 2.1). The naive minmax approach maintains this cost equivalence, since the maximum loss occurs at interval boundaries. For minmax with smoothness constraints through regularization, our alternating gradient descent-ascent updates for $f$ and $f'$ double the computational overhead. The pseudo-label approach requires training $k$ hypotheses from $\widetilde{\mathcal{F}}_\eta$ before generating labels, resulting in $(k + 1)$ times the base cost - typically manageable given efficient regression training.

## G.2   Experiment setup

Following prior work [Cheng et al., 2023a], we conducted experiments on five public datasets from the UCI Machine Learning Repository: Abalone, Airfoil, Concrete, Housing, and Power Plant. Since these datasets are originally regression tasks with single target values, we transformed them into datasets with interval targets (described shortly). Dataset statistics are provided in Section H. For the experimental setup, we used the same configuration as [Cheng et al., 2023a]: the model architecture is a MLP with hidden layers of sizes 10, 20, and 30. We trained the models using the Adam optimizer with a learning rate of 0.001 and a batch size of 512 for 1000 epochs.

**Interval Data Generation Methodology.**   We propose a general approach for generating interval data for each target value $y$. This method depends on two factors: the interval size $q \in [0, \infty]$ and the interval location $p \in [0, 1]$. The interval is then defined as $[l, u] = [y - pq, y + (1 - p)q]$. When $p = 0$, the target value $y$ is at the lower boundary of the interval whereas $p = 1$ places $y$ at the upper boundary. In this work, we consider $q$ and $p$ to be generated from uniform distributions over specified ranges. The prior interval generation method in Cheng et al. [2023a] could be seen as a special case of our approach when $q \sim \text{Uniform}[0, q_{\max}]$ and $p \sim \text{Uniform}[0, 1]$.

## G.3   Results

**Which method works best in the uniform setting?**   We begin by evaluating methods in the uniform interval setting described in prior work [Cheng et al., 2023a], where the interval size $q \sim \text{Uniform}[0, q_{\max}]$ and the location of the interval $p \sim \text{Uniform}[0, 1]$. For each dataset, we set $q_{\max}$ to be approximately equal to the range of the target values, $y_{\max} - y_{\min}$. Specifically, we set $q_{\max} = 30$ (Abalone), 30 (Airfoil), 90 (Concrete), 120 (Housing), and 90 (Power Plant). Our findings indicate that the PL (mean) method performs best in this uniform setting, with PL (max) and the projection method ranking second and third, respectively (Table 2). Given the superior performance of PL (mean), we conducted an ablation study to better understand its effectiveness. We explored the impact of varying the number of hypotheses $k$ and compared it with an ensemble baseline that combines pseudo-labels *before* using them to train the model, for which we still find

that PL (mean) still performs better (Appendix K).

**What about other interval settings?** We conducted more detailed experiments to investigate which factors impact the performance of each method. Specifically, we varied the interval size $q$ and the interval location $p$ by 1) varying $q_{\max}$, 2) varying $q_{\min}$, 3) varying $p$ with three settings designed to position the true value $y$ at: i) *only* one boundary of the interval, ii) *both* boundaries of the interval, iii) the middle of the interval. Full details are provided in Appendix I. We found that: (1) All methods are quite robust to changes in the interval size, except for the Minmax method, whose performance decreases significantly as the interval size increases. This is consistent with our insights from the proof of 5.4), (2) The location of the true value $y$ can have a large impact on performance; specifically, the Minmax method performs better when $y$ is close to the middle of the interval. One explanation is that Minmax is equivalent to supervised learning with the midpoint of the interval (Corollary 5.2). Conversely, the other methods perform better when $y$ is close to *both* boundaries of the interval but not when $y$ is close to *only* one boundary. Finally, we conclude that if we only know that the interval size is large, it is better to use the PL (pseudo-labeling). However, if we know the true value $y$ is close to the middle of the interval, then the Minmax method is more preferable.

### G.4 Connection to our theoretical analysis

To validate our theoretical findings in practice, we conducted experiments designed to test whether our theory holds under empirical conditions. Recall that our main result (Theorem 3.6) states that if a hypothesis $f$ approximately lies within the intervals ($f \in \widetilde{\mathcal{F}}_\eta$) and is smooth, then $f$ will lie within intervals smaller than the original ones. To control the smoothness of our hypothesis, we utilize a Lipschitz MLP, which is an MLP augmented with spectral normalization layers [Miyato et al., 2018]. The normalization ensures that the Lipschitz constant of the MLP is less than $1$. We then scale the output of the MLP by a constant factor $m$ to ensure that the Lipschitz constant of the hypothesis is less than $m$.

**Test performance** First, we plot the test Mean Absolute Error (MAE) of the Lipschitz MLP with the projection objective, compared with the test MAE of the standard MLP (Figure 8 (Top)). We found that, with the right level of smoothness, Lipschitz MLP can achieve better performance than the standard MLP. When the Lipschitz constant is very small, the performance is poor for all datasets. However, performance improves as the Lipschitz constant increases. We observe that the optimal Lipschitz constant is always larger than the Lipschitz constant estimated from the training set (vertical line). For some datasets, performance degrades when the Lipschitz constant becomes too large. This aligns with our insight from Theorem E.5, which suggests that we can improve the error bound by ensuring that the hypothesis class is as smooth as possible (smaller $m$ so that $I_\eta(x)$ is small) while still containing a good hypothesis (i.e., low OPT). Nevertheless, we do not need to know the Lipschitz constant of the dataset and can treat it as a tunable hyperparameter in practice.

**Reduced interval size** Second, we determine whether the intervals, within which our hypothesis $f \in \widetilde{\mathcal{F}}_0$ lies, are smaller than the original intervals. Recall that the original intervals are given by $[l, u]$, and our theorem suggests that they would reduce to $I_\eta(x) = [\tilde{l}_{\mathcal{D} \to x}^{(m)} - r_\eta(x), \tilde{u}_{\mathcal{D} \to x}^{(m)} + s_\eta(x)]$. While we can use a Monte Carlo approximation to estimate $I_\eta(x)$, it does not take into account the hypothesis class $\mathcal{F}$. Instead, we approximate $I_\eta(x)$ using samples of hypotheses from $\widetilde{\mathcal{F}}_0$ by proceeding as follows: 1) We train 10 models with the projection objective, each from different random initializations (denoted by $f_1, \ldots, f_{10}$), 2) For each $x$, we approximate the reduced interval using the minimum and maximum values of the outputs from these models, given by $[\min_i f_i(x), \max_i f_i(x)]$. We set $m \in \{0.1, 0.1 \times 2^1, \ldots, 0.1 \times 2^{13}\}$ and consider a uniform interval setting with $q_{\max} = 90$. As expected, when the hypothesis becomes smoother, we observe that the average interval size decreases (Figure 8 (Bottom)). Moreover, we found that even when the Lipschitz constant is much larger than the value estimated from the data (vertical line), the average reduced interval size remains significantly smaller than the original interval (which is $45$ since $q_{\max} = 90$). We also observe that the average interval sizes from the standard MLPs are smaller than the original values.

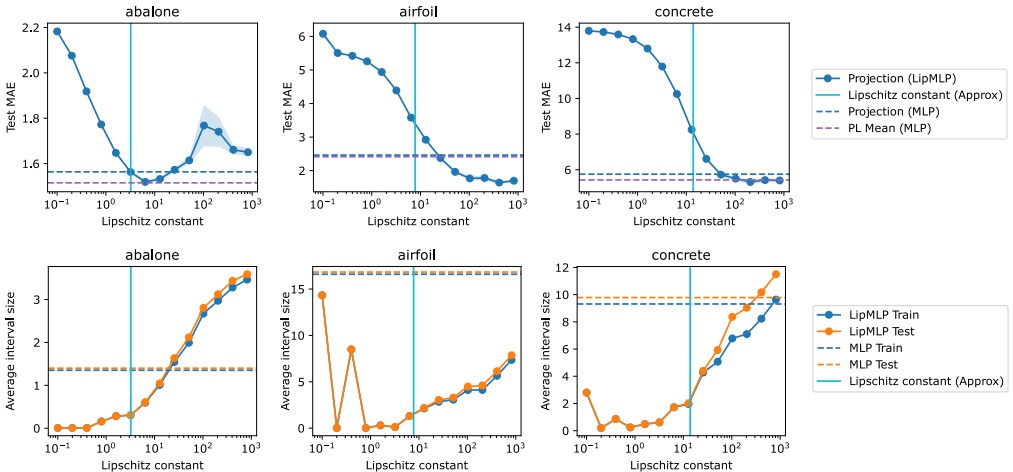

Figure 8: Test MAE of the projection method with Lipschitz MLP using different values of the Lipschitz constant. The vertical line is the Lipschitz constant approximated from the training set. (Top) The dashed horizontal lines are the test MAE of PL (Mean) and Projection approach with a standard MLP. (Bottom) Approximated interval size $I_\eta(x)$ for Lipschitz MLP with a different value of Lipschitz constant $m$. The dashed horizontal lines are the values from standard (non-Lipschitz) MLP. The figures for all datasets are in Appendix J.

## H   Dataset Statistics

The datasets are from the UCI Machine learning repository [Nash et al., 1994, Brooks et al., 1989, Yeh, 1998, Tfekci and Kaya, 2014] with Creative Commons Attribution 4.0 International (CC BY 4.0) license. We provide the statistics of the datasets including the number of data points, the number of features, the minimum and maximum values of the target value and the approximated Lipschitz constant in Table 3. The Lipschitz constant here is approximated by calculating the proportion $\frac{|y-y'|}{\|x-x'\|}$ for all pairs of data points then the value is given by the 95th percentiles of these proportions. We perform this procedure to avoid the outliers which have a size of around two orders of magnitude bigger than the 95th percentile value (Figure 9). This allows us to approximate the level of smoothness that does appear in the dataset rather than use the maximum Lipschitz constant. One could also think of this as a probabilistic Lipschitz value rather than the classical notion [Urner and Ben-David, 2013].

| Dataset | # data points | # features | [y min, y max] | Lipschitz constant |
|---------|---------------|------------|----------------|--------------------|
| Abalone | 4177 | 10 | [1,29] | 3.23 |
| Airfoil | 1503 | 5 | [103, 141] | 7.75 |
| Concrete | 1030 | 8 | [2,83] | 13.8 |
| Housing | 414 | 6 | [7, 118] | 11.68 |
| Power plant | 9568 | 4 | [420,496] | 14.18 |

Table 3: Dataset statistics.

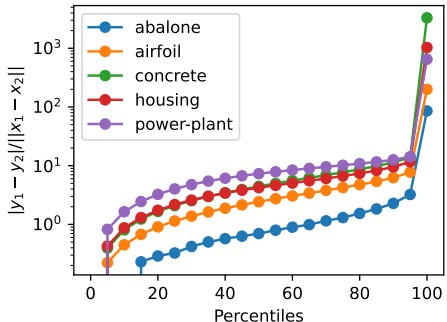

Figure 9: The value of $\frac{|y-y'|}{\|x-x'\|}$ by percentiles. We use the 95th percentile of this value as an approximated Lipschitz constant for each dataset.

# I Impacts of the interval size and interval location

## I.1 Impact of the interval size

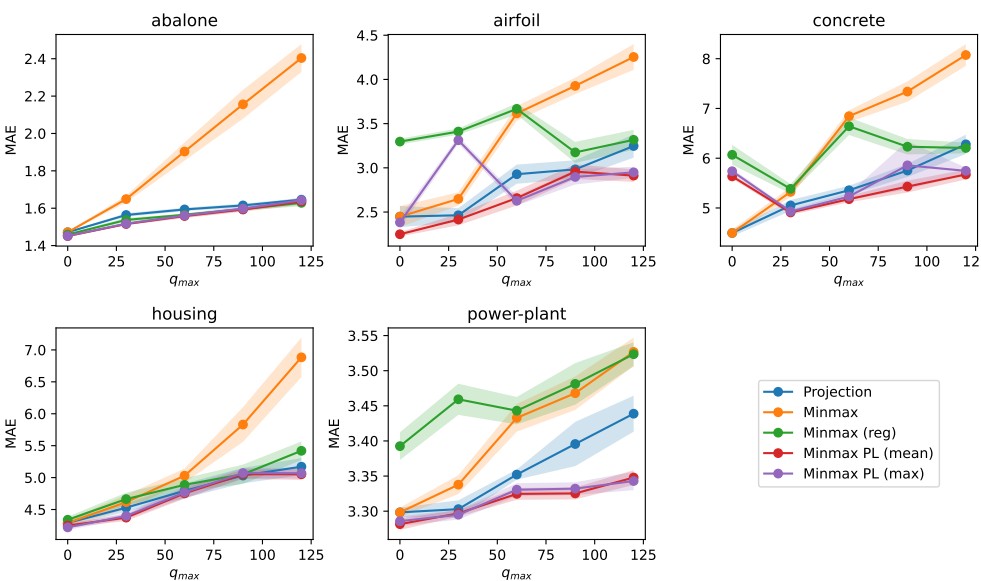

Figure 10: Test MAE when varying the maximum interval size $q_{\max} \in \{0, 30, 60, 90, 120\}$ while $q_{\min} = 0$.

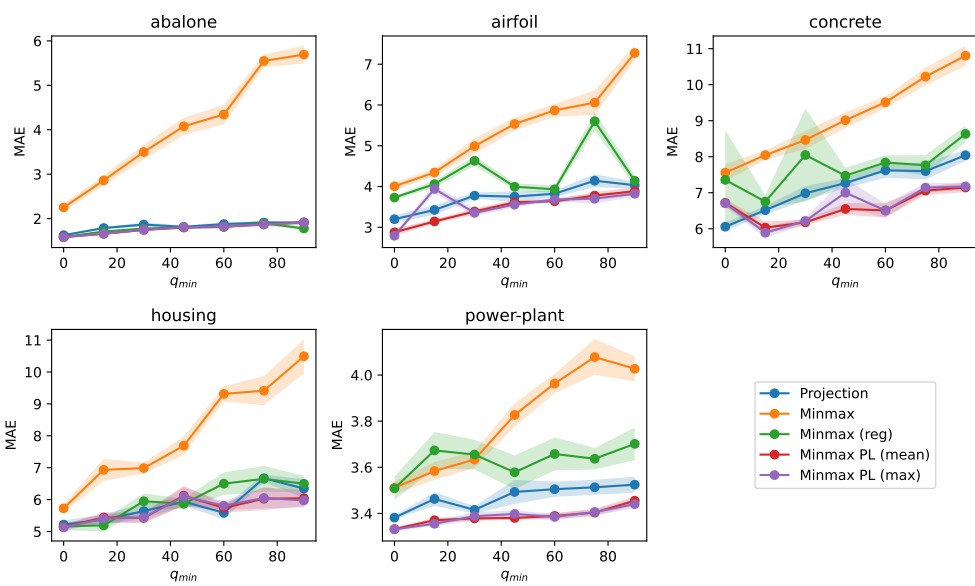

Figure 11: Test MAE when varying the minimum interval size $q_{\min} \in \{0, 15, 30, 45, 60, 75, 90\}$ while $q_{\max} = 90$.

We want to investigate the impact of interval size on the performance of the proposed methods. Intuitively, a smaller interval would make the problem easier. In the extreme case when the interval size is zero, we recover the supervised learning setting. Here, we assume that the interval location $p$ is still drawn uniformly from $[0, 1]$ and we consider two experiments. First, we vary the maximum interval size $q_{\max} \in \{0, 30, 60, 90, 120\}$ while keeping the minimum interval size $q_{\min} = 0$. As

expected, a larger maximum interval size leads to the drop in test performance across the boards (Figure 10). Second, we vary the minimum inter val size $q_{\min} \in \{0, 15, 30, 45, 60, 75, 90\}$ while keeping $q_{\max}$ fixed at 90. We can see that the test performance also decreases for all methods as we increase the minimum interval size (Figure 11). Notably, the standard minmax approach is highly sensitive to the interval size where its performance degrades significantly much more than other approaches in both experiments. This is due to the nature of the approach that wants to minimize the loss with respect to the worst-case label, as we have a larger interval, these worst-case labels can be much stronger and may not represent the property of the true labels anymore. On the other hand, our other minmax approaches and the projection approach are more robust to the change in the minimum interval size and the error only went up slightly for both experiments.

## I.2 Impact of the interval location

**When $y$ is more likely to be on one side of the interval (vary $p_{\min}$)**

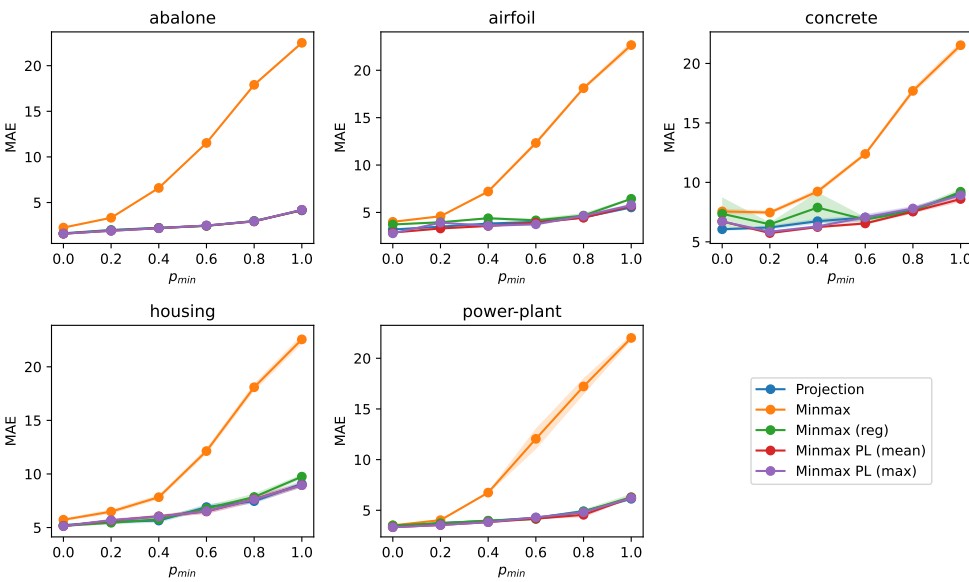

Figure 12: Test MAE when varying the minimum interval location $p_{\min} \in \{0, 0.2, 0.4, 0.6, 0.8, 1\}$. In this case, when $p_{\min} = 0$ we have the uniform interval setting while when $p_{\min} = 1$, $y$ true always lie on the upper bound of the intervals.

In the previous settings, we assume that the location of the interval $p$ is drawn uniformly from $U[0, 1]$, that is, when $y$ true is equally likely to be located at anywhere on the intervals. Here, we explore what would happen when it is not the case. We assume that we fixed $q_{\min} = 0$, $q_{\max} = 90$ and consider three scenarios. First, we consider when $y$ is more likely to be on one side of the interval. Here, we consider when $p \sim U[p_{\min}, 1]$ where $p_{\min} \in \{0, 0.2, 0.4, 0.6, 0.8, 1\}$ (Figure 12). In this case, when $p_{\min} = 0$ we have the uniform interval setting while when $p_{\min} = 1$, $y$ true always lies on the upper bound of the intervals. We can see that the test MAE of all approaches increases as $p_{\min}$ is larger. Again, the minmax approach performs much worse than others. One explanation for this is that the minmax with respect to. the label would encourage the model to be close to the middle point of each interval (Corollary 5.2). However, the the $y$ true is far away from the midpoint leads to his phenomenon. We also provide the test MAE with no minmax approach for better visualization (Figure 13)

**When $y$ true is more likely to be in the middle of the interval**

Second, we consider when $y$ true is more likely to be in the middle of the interval ($p$ is close to $0.5$). We capture this setting by considering $p \sim U[0.5 - c, 0.5 + c]$ for $c \in \{0, 0.1, 0.2, 0.3, 0.4, 0.5\}$ (Figure 14). Intuitively, when $c = 0$, the true $y$ is always in the middle of the interval and when $c = 0.5$, we recover the uniform interval setting. In contrast to the first setting, we can see that the minmax approach performs the best in this setting for a small value of $c$. Again, this is perhaps due

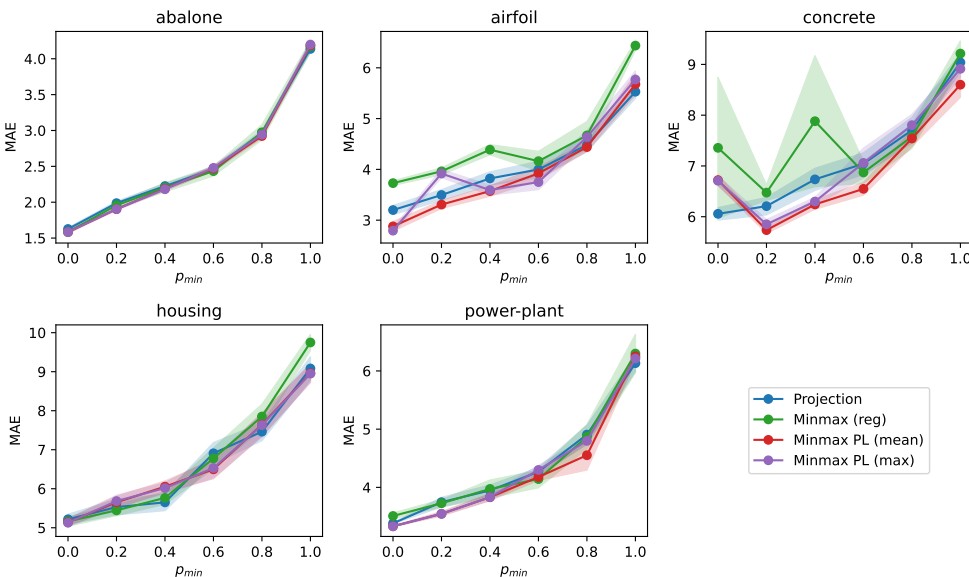

Figure 13: Test MAE when varying the minimum interval location $p_{\min} \in \{0, 0.2, 0.4, 0.6, 0.8, 1\}$. In this case, when $p_{\min} = 0$ we have the uniform interval setting while when $p_{\min} = 1$, $y$ true always lies on the upper bound of the intervals.(no minmax approach)

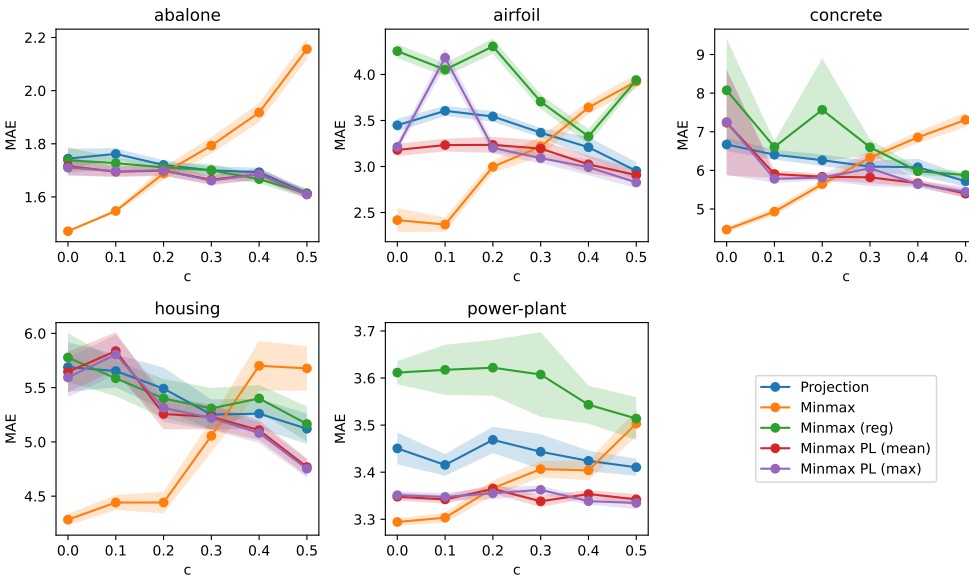

Figure 14: Test MAE when varying the interval location, $p \sim U[0.5 - c, 0.5 + c]$ for $c \in \{0, 0.1, 0.2, 0.3, 0.4, 0.5\}$. When $c = 0$, the true $y$ is always in the middle of the interval and when $c = 0.5$, we recover the uniform interval setting.

to the nature of the minmax approach mentioned earlier which encourages the prediction to be close to the middle point of the interval, for which, in this case, close to the $y$ true. Remarkably, minmax performs better until $c = 0.2$ which corresponds to $p \sim [0.3, 0.7]$ which is a reasonable location of $y$ true in practice. However, when $c$ is large we would recover the uniform interval setting and the minmax would go back to becoming the worst-performer. On the other hand, the performance of other approaches is better as $c$ is larger, that is when $y$ true is more spread out across the interval.

**When $y$ is more likely to be on either side of the interval**

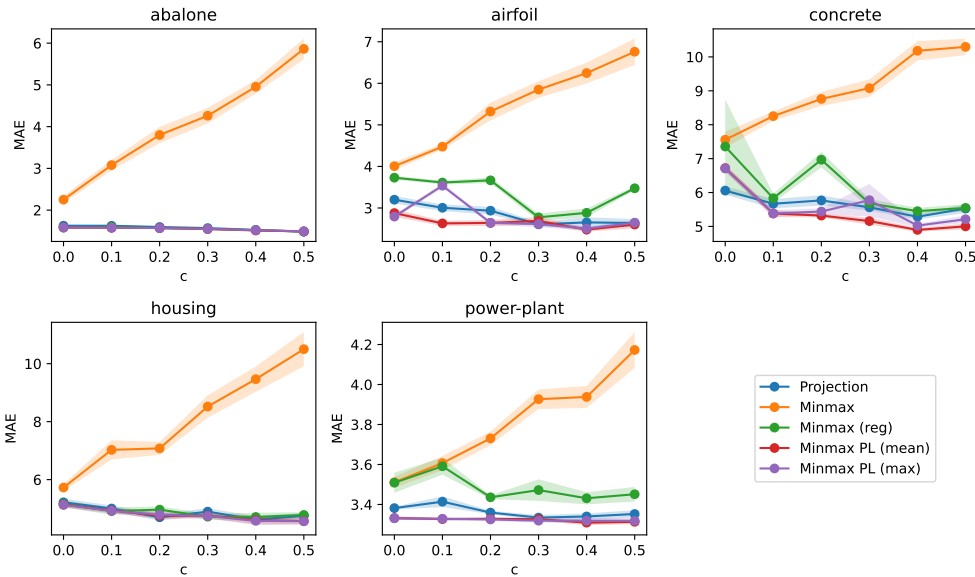

Figure 15: Test MAE when varying the interval location, when $p$ is drawn uniformly from $[0, 0.5 - c] \cup [0.5 + c, 1]$ when $c \in \{0, 0.1, 0.2, 0.3, 0.4, 0.5\}$. Here, when $c = 0$ we have the uniform interval setting while when $c = 0.5$, $y$ true is either on the upper or the lower bound of the intervals.

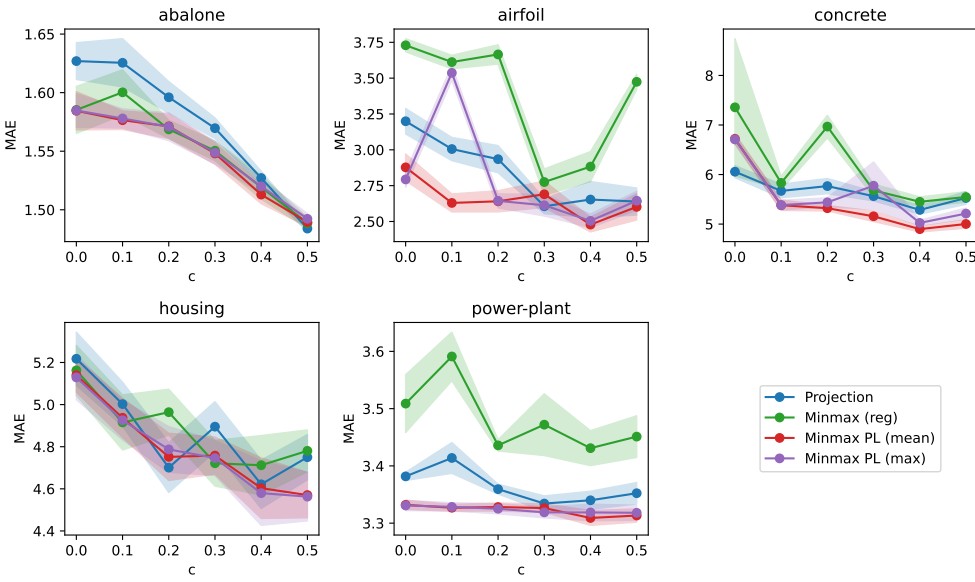

Figure 16: Test MAE when varying the interval location, when $p$ is drawn uniformly from $[0, 0.5 - c] \cup [0.5 + c, 1]$ when $c \in \{0, 0.1, 0.2, 0.3, 0.4, 0.5\}$. Here, when $c = 0$ we have the uniform interval setting while when $c = 0.5$, $y$ true is either on the upper or the lower bound of the intervals.(no minmax approach)

Finally, we consider when $y$ is more likely to be on either side of the interval where $p$ is drawn uniformly from $[0, 0.5 - c] \cup [0.5 + c, 1]$ when $c \in \{0, 0.1, 0.2, 0.3, 0.4, 0.5\}$. Here, when $c = 0$ we have the uniform interval setting while when $c = 0.5$, $y$ true is either on the upper or the lower bound of the intervals. We found that as $c$ is larger where the $y$ true is more likely to be near either of the boundaries, the minmax performance drop significantly (Figure 15). However, we found that the per-

formance of other approaches increases (Figure 16). This is in contrast to the first setting where we see that when $y$ is more likely to be near only one side of the boundary, the performance drops remarkably.

Overall, from these experiments, we may conclude that for all approaches apart from the original minmax with respect to. labels, having $y$ true that lies near both of the boundaries of the interval are beneficial to the test performance and lying on both sides is crucial.

### I.3 Large Ambiguity degree setting

We consider a setting with large ambiguity degree where $q \sim \mathrm{Uniform}[q_{\min}, 90]$ when $q_{\min} \in \{30, 60, 90\}$ and $p \sim \mathrm{Uniform}[0.5 - c, 0.5 + c]$ when $c \in \{0, 0.1, 0.2, 0.3, 0.4, 0.5\}$. Here as $c$ is smaller, $y$ true would be located near the middle point of the interval while as $c$ is larger, we would recover the uniform setting. These settings have a large ambiguity degree since when $q_{\min} > 0$, interval size can't be arbitrarily small and $[p_{\min}, p_{\max}] \subset [0, 1]$ implies that true y would not lie at the boundary of the constructed interval. As a result, the intersection of all possible intervals would no longer be just $\{y\}$ anymore which leads to the ambiguity degree of $1$. We found that there is no single method that always performs well on every interval setting. The Minmax is the best performing method for all $c \leq 0.3$ while when $c > 0.3$ the best-performing approaches are either PL (mean) or PL (max) (Figure 17).

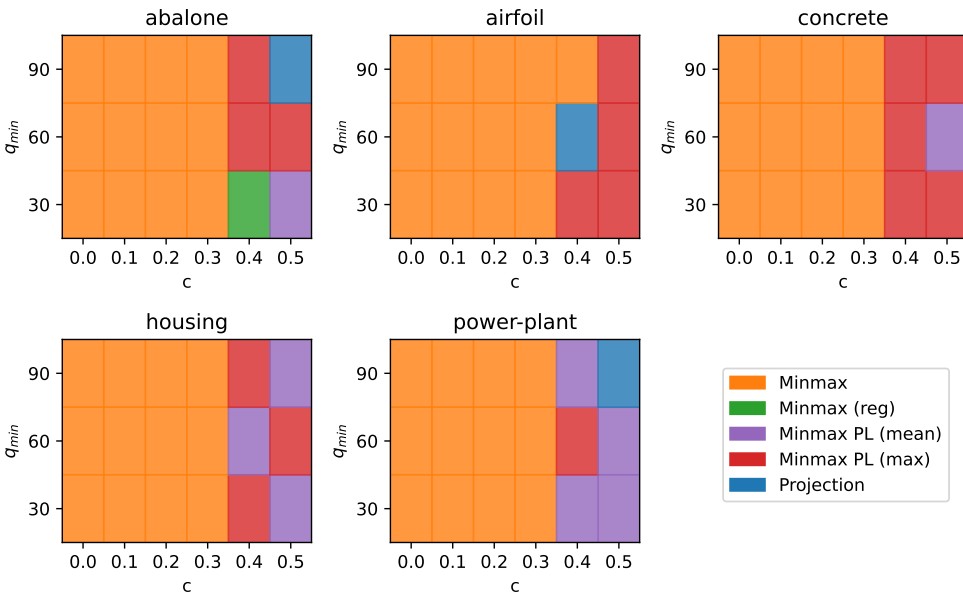

Figure 17: The best performing approach for each $c$ and $q_{\min}$

## I.4 Interval padding experiment

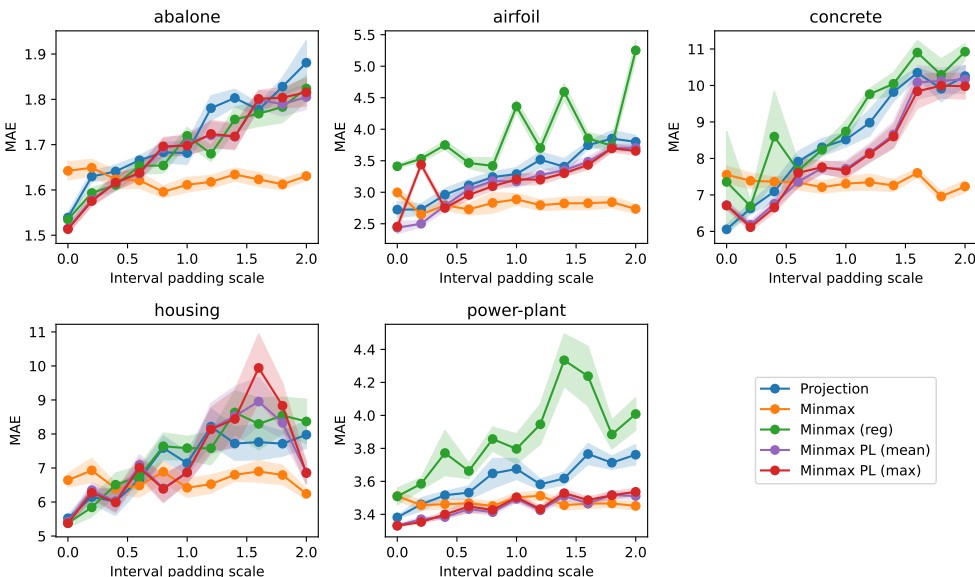

Figure 18: Interval padding experiment

From above, we found that the Minmax approach performs better when $y$ true is close to the middle point of the interval, but performs worse in the uniform interval setting when $p \sim \text{Uniform}[0, 1]$. In this experiment, we start with the uniform interval setting and add padding to the original interval as a factor of the interval size. Formally, for an original interval $[l, u]$ of size $q = u - l$, we have a new interval $[l - sq, u + sq]$ when $s > 0$ is a scale parameter. By doing this, $y$ true would be *proportionally* closer to the midpoint of the new interval, but distancewise is the same. We found that as we add the padding, the performance of other approaches decreases significantly and gets worse than the performance of the Minmax when the scale is $0.5$ (when the padded interval is twice the size of the original interval) while the performance of Minmax is about the same. This shows that a redundant interval (padding) can harm the performance of the proposed approaches except Minmax and our result that interval location $p$ can have a large impact on the performance is still applicable to this padding setting.

# J    Interval size and test performance of LipMLP

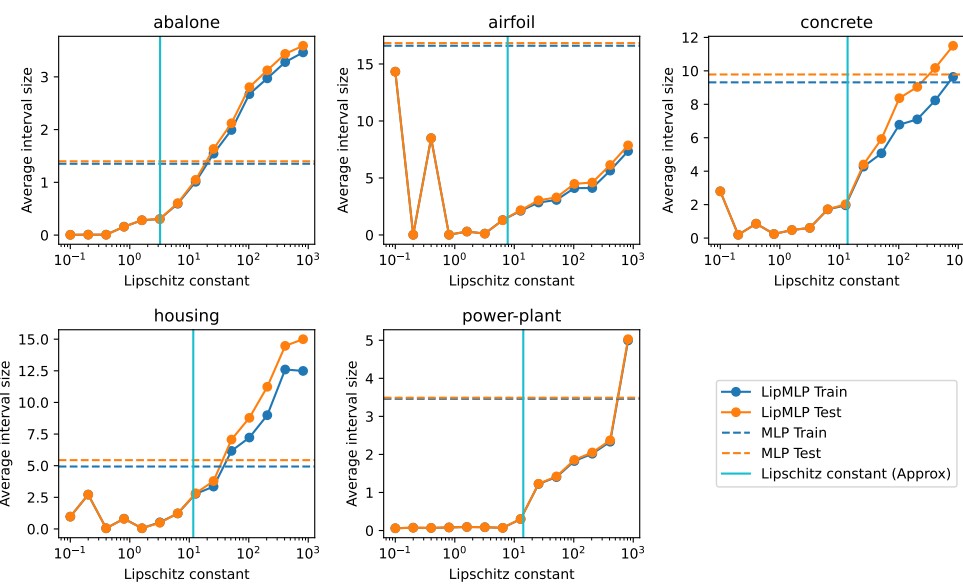

Figure 19: Approximated interval size $I_\eta(x)$ for Lipschitz MLP with a different value of Lipschitz constant $m$. The dashed horizontal lines are the values from standard MLP.

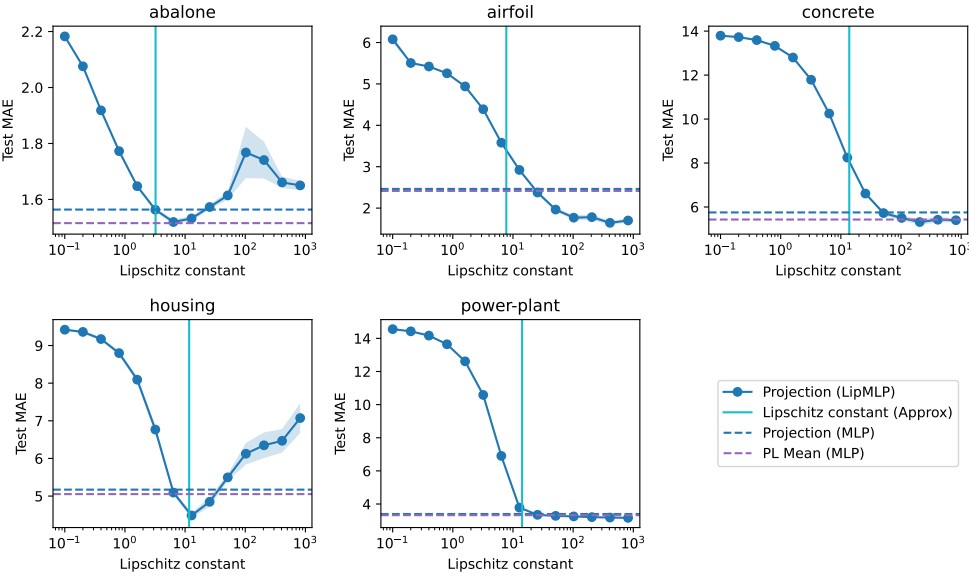

Figure 20: Test MAE of the projection method with Lipschitz MLP with different values of Lipschitz constant. The vertical line is the Lipschitz constant approximated from the training set. The dashed horizontal lines are the test MAE of PL (Mean) and Projection approach with a standard MLP. The optimal Lipschitz constant balances the trade-off between constraining the hypothesis class and maintaining enough capacity to achieve low error.

## K   Ablation for PL (mean)

Since PL (mean) is the best-performing approach in the uniform interval setting, we also performed an ablation study to improve our understanding of this method. First, we explore the impact of the number of hypotheses $k$ used to represent $\widetilde{\mathcal{F}}_0$. We found that for every dataset, as $k$ is larger, the test MAE becomes smaller. While we use $k = 5$ for all PL experiments, this ablation suggests that we can increase $k$ to get better performance at the cost of more computation.

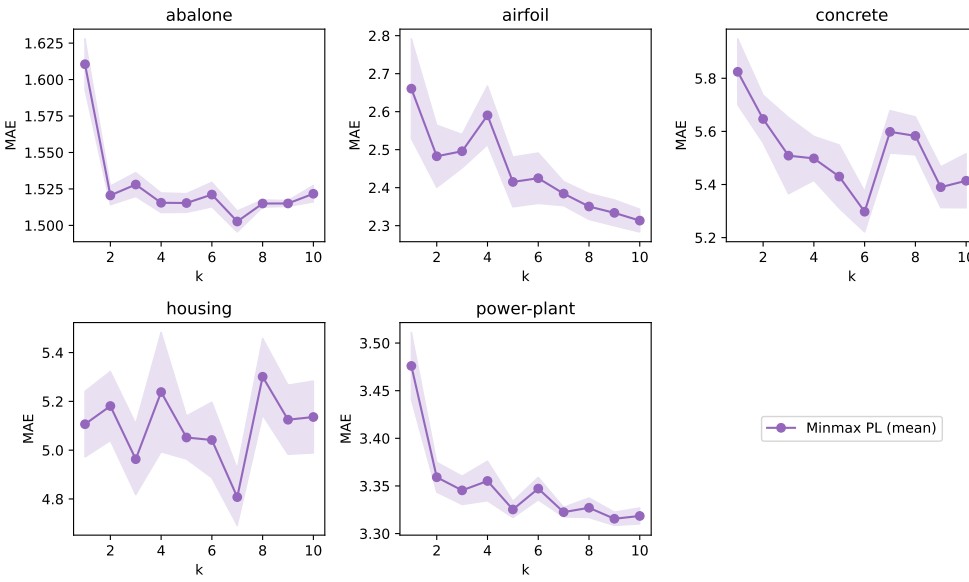

Figure 21: Test MAE for PL (mean) with different number of hypotheses $k$ used to represent $\widetilde{\mathcal{F}}_0$. For almost every dataset, the test MAE decreases as $k$ is larger.

Second, we also compare PL (mean) with a natural ensemble baseline where we combine pseudo labels by averaging them first and then train a model with respect to. the averaged labels. In particular, the objective for the ensemble baseline is given by

$$\min_f \sum_{i=1}^{n} \ell(f(x_i), \sum_{j=1}^{k} f_j(x_i)). \tag{134}$$

We found that PL (mean) still performs better than this baseline on 2 out of 5 datasets while the other 3 datasets are similar.

|  | Abalone | Airfoil | Concrete | Housing | Power-plant |
|---|---|---|---|---|---|
| PL (mean) | $1.52_{0.01}$ | $2.42_{0.07}$ | $5.43_{0.12}$ | $5.05_{0.09}$ | $3.33_{0.01}$ |
| PL ensemble baseline | $1.51_{0.01}$ | $3.3_{0.04}$ | $5.57_{0.19}$ | $5.06_{0.08}$ | $3.32_{0.01}$ |

Table 4: Test Mean Absolute Error (MAE) and the standard error (over 10 random seeds) for PL (Mean) and a PL ensemble baseline

## L   Additional experiments on the tabular data benchmark

The main takeaway from our theoretical analysis is that an appropriate level of smoothness can lead to a performance gain. In addition to our experiments on the UCI datasets, we also tested this on 18 additional regression tasks from a tabular data benchmark [Grinsztajn et al., 2022]. To ensure

that the MAEs of different datasets are comparable, we used z-score rescaling on the target values of each dataset so that the standard deviation was 100. We only used the training datasets to infer the rescaling parameters. To generate the interval targets, we used the proposed algorithm with $q_{\min} = 0, q_{\max} = 50, p_{\min} = 0$, and $p_{\max} = 1$. In our experiment, we compared MLP with LipMLP using different values for the Lipschitz constants, where $m \in \{1, 4, 16, 64, 256, 1024\}$. We used a validation dataset to select the best hyperparameters, which included the learning rate for MLP (from $\{0.01, 0.001, 0.0001, 0.00001\}$) and both the learning rate and the Lipschitz constant for LipMLP. We provide the test MAE with standard error over 5 random seeds for both methods in Table 1. We bolded the result whenever the mean + standard error (ste) of one method was lower than the mean - ste of the other method. We found that on almost every dataset (apart from GPU), LipMLP performed better than or at least on par with MLP. This extensive improvement demonstrates suggests that determining the right level of smoothness is a simple yet effective method for enhancing learning with interval targets.

