# OpenReview forum: "Learning from Interval Targets"
_NeurIPS.cc/2025/Conference — NeurIPS 2025 poster_

### Official Review · Reviewer_N93J · 2025-06-26

**Clarity:** 2
**Significance:** 3
**Originality:** 3
**Rating:** 4
**Confidence:** 4

**Summary:**

This paper addresses a weakly supervised regression problem where, instead of exact labels, only upper and lower bounds of the target values are provided. This problem is easier to collect labels than the traditional supervised setting. While this setting has been explored in prior work, existing approaches typically rely on strong theoretical assumptions. In contrast, this paper adopts more relaxed assumptions and offers several theoretical results that shed light on the problem, particularly regarding the influence of function class smoothness. The authors also explore a variety of objective functions suitable for learning from interval-based supervision and empirically evaluate their performance through comprehensive experiments.

**Questions:**

1. Clarity and Self-Containment:
Please consider addressing the weaknesses I noted regarding the organization and self-containment of the paper. In particular, key experimental details and the best-performing objective function (Equation 134) are only described in the appendix, which hinders comprehension of the main results. If I have misunderstood or overlooked any explanations already present in the main text, I would appreciate clarification.
2. Comparison with Noisy Label Regression:
I believe it would be quite interesting to compare this interval-based framework with the standard setting of regression from noisy ground truth labels. Specifically, how would the proposed method perform relative to learning from noisy scalar labels, and are there scenarios where interval targets offer a clear advantage? I would be interested in hearing your perspective on this.
3. Fair Comparison with PL (Mean) Approach:
While the PL (mean) approach is shown to be highly effective, it requires training K+1 regressors, which introduces a significant computational cost. For a fairer comparison in terms of training time, it would be helpful to evaluate the performance of an ensemble of K+1 models using the minmax or projection-based objectives. Have you explored this comparison, or do you have any expectations about how such ensembles might perform?

**Ethical Concerns:**

["NO or VERY MINOR ethics concerns only"]

**Final Justification:**

After reading the rebuttal, my score remains unchanged, as there are still concerns regarding the clarity of the paper. I am not entirely confident that the authors can adequately revise it to address these issues. In particular, the explanations of the experiments remain unclear, and it is difficult to fully grasp what the proposed method entails. I understood that the key idea is to control the Lipschitzness and use the existing loss is acceptable (using PL (mean) can be effective in many cases). That said, I do appreciate the effort to generalize beyond Cheng et al., and I find the theoretical analysis to be interesting and a meaningful contribution to advancing the understanding of learning from interval targets. Overall, I feel this paper is at a borderline level.

**Limitations:**

yes (but limitation was discussed in Appendix B)

**Quality:**

3

**Strengths And Weaknesses:**

Strengths
1. This paper addresses an important and compelling problem setting, proposing a practical algorithm that is flexible with respect to model choice, easy to implement, and empirically effective.
2. The theoretical analysis is thorough and well-presented. The authors provide clear interpretations that help readers grasp the core insights of the theoretical results.
3. For a natural baseline in this setting is to regress the midpoint of the lower and upper bounds; this paper demonstrates that such an approach can be overly pessimistic in terms of the worst-case (min-max) loss. Building on this insight, the authors propose a more optimistic and effective alternative. This result is, in my view, a significant contribution to the field.

Weaknesses
1. While I appreciate the overall contribution of the paper, I found that it is not fully self-contained, and substantial reorganization may be needed to meet the standards of a top-tier conference paper. Although the theoretical analysis is clearly presented in the main text, the algorithmic and experimental sections are difficult to follow without frequent reference to the appendix. Notably, the best-performing objective function used in the experiments is only described in the appendix (Equation 134), which makes it challenging for readers to fully understand or assess the empirical results based on the main paper alone.
2. The paper reports results on only five UCI datasets, whereas Cheng et al. (2023a)—which the paper claims to follow—uses six datasets, including AgeDB, IMDB-WIKI, and STS-B. Including a more diverse and challenging set of datasets would help better demonstrate the generalizability of the proposed method.
3. The experimental comparison does not include strong baselines from related prior work, such as the LM loss introduced in Cheng et al. (2023a). Including such methods would strengthen the empirical evaluation and clarify whether the proposed approach offers a clear improvement over existing techniques.

---

> ### Author Response · Authors · 2025-08-01
>
> Thank you for the thoughtful feedback. Please find our individual responses below.
>
> > While I appreciate the overall contribution of the paper, I found that it is not fully self-contained, and substantial reorganization may be needed to meet the standards of a top-tier conference paper. Although the theoretical analysis is clearly presented in the main text, the algorithmic and experimental sections are difficult to follow without frequent reference to the appendix. Notably, the best-performing objective function used in the experiments is only described in the appendix (Equation 134), which makes it challenging for readers to fully understand or assess the empirical results based on the main paper alone.
>
> Our original intention was to prioritize the theoretical development in the main text due to strict page limits, but we recognize that this came at the cost of clarity.
>
> Based on your feedback, we will revise the manuscript to improve its self-containment. Specifically, we will move the definition of our primary objective function (formerly Eq. 134) and other key experimental details from the appendix into the main body of the paper.
>
>
>
> > The paper reports results on only five UCI datasets, whereas Cheng et al. (2023a)—which the paper claims to follow—uses six datasets, including AgeDB, IMDB-WIKI, and STS-B. Including a more diverse and challenging set of datasets would help better demonstrate the generalizability of the proposed method.
>
>
>
> We have performed an extensive evaluation on 18 datasets, but unfortunately these results had to be moved the Appendix due to space constraints. As detailed in Appendix L (Table 4, page 45), we evaluated our method on 18 diverse tabular datasets from the benchmark suite of Grinsztajn et al. [1]. **Our LipMLP significantly outperforms the standard MLP on 14 of the 18 datasets**, with similar or competitive performance on the rest. This provides strong evidence for the generalizability of our approach, and we will revise the main text to better highlight these extensive results.
>
> [1] Grinsztajn, Léo, Edouard Oyallon, and Gaël Varoquaux. "Why do tree-based models still outperform deep learning on typical tabular data?." Advances in neural information processing systems 35 (2022): 507-520.
>
>
> > The experimental comparison does not include strong baselines from related prior work, such as the LM loss introduced in Cheng et al. (2023a). Including such methods would strengthen the empirical evaluation and clarify whether the proposed approach offers a clear improvement over existing techniques.
>
> The LM loss from [Cheng et al.] is a specific instance of our more general projection loss framework when the loss function is set to l1​. Therefore, our "Projection MLP" baseline in the experiments is, in fact, equivalent to the LM loss baseline the reviewer requested. Our results show that our proposed LipMLP consistently outperforms this strong baseline on numerous datasets. We will revise the paper to make this equivalence explicit.
>
>
>
>
>
> > Comparison with Noisy Label Regression: I believe it would be quite interesting to compare this interval-based framework with the standard setting of regression from noisy ground truth labels. Specifically, how would the proposed method perform relative to learning from noisy scalar labels, and are there scenarios where interval targets offer a clear advantage? I would be interested in hearing your perspective on this.
>
> Please see the general response.
>
>
>
> > Fair Comparison with PL (Mean) Approach: While the PL (mean) approach is shown to be highly effective, it requires training K+1 regressors, which introduces a significant computational cost. For a fairer comparison in terms of training time, it would be helpful to evaluate the performance of an ensemble of K+1 models using the minmax or projection-based objectives. Have you explored this comparison, or do you have any expectations about how such ensembles might perform?
>
> All methods have a pretty low computation anyway since all of them are at a constant factor of a standard liner regression. Therefore, we think the difference in running time is not significant.
>
> The performance of an ensemble is also interesting. In Appendix K, we compare PL(mean) with a natural ensemble baseline where we combine pseudo labels by averaging them first and then train a model with respect to the averaged labels. We found that PL (mean) still performs better than this baseline on 2 out of 5 datasets while the other 3 datasets are similar.

---

> > ### Comment · Reviewer_N93J · 2025-08-03
> > **Thank you for the rebuttal**
> >
> > I appreciate the authors' response to my concerns in detail.
> >
> > 1. I understand that the page limit can be challenging. if this paper is accepted, I strongly recommend improving the clarity of the experimental results section in the main body. For example, Figure 7 is difficult to interpret based solely on the text, e.g., terms like PL (mean) appear without any explanation, Lipschitz MLP is not explained, and why this figure is introduced.
> > 2. Thank you for clarifying the inclusion of the LM loss in Cheng et al. it would be very helpful to state this explicitly in the paper.
> > 3.  In the main body, there is also no explanation of LipMLP (Figure 7). What exactly is this method? From my reading of the appendix, my understanding is that it applies spectral normalization [Miyato et al., 2018], scales it by the factor $m$, and then uses the projection loss proposed by Cheng et al. as the objective function. Is this interpretation correct?

---

> > > ### Author Response · Authors · 2025-08-04
> > >
> > > Thank you for the constructive feedback. We will incorporate all your suggestions into the camera-ready version. To confirm, your interpretation of LipMLP is correct, and we will add a clear definition to the main body. We appreciate your help in strengthening the paper.

---

### Official Review · Reviewer_U3a1 · 2025-06-30

**Clarity:** 4
**Significance:** 3
**Originality:** 3
**Rating:** 5
**Confidence:** 3

**Summary:**

The paper extends regression with interval-valued supervision by dropping the restrictive “ambiguity-degree” assumption and replacing it with a global Lipschitz-smoothness condition on the hypothesis class. It derives new generalization bounds for the standard projection loss and introduces a complementary min-max surrogate, then shows that enforcing an appropriate Lipschitz constant can reduce test MAE on a suite of tabular datasets.

Please note that the following represents my initial impressions of the paper, and I am open to discussion. I welcome any corrections to potential misunderstandings.

**Questions:**

My actionable requests already appear in the Weaknesses section; please address them point-by-point. In addition, I highlight a few items here:

- Could you extend Fig.7's $m$ sweep until MAE stats rising? If MAE does not raise, please add discussion why it is still compatible with what the theory expects.
- Could you possibly release an anonymous sample code so that I can evaluate parameter sensitivity of $m$, learning rate, weight-decay, overfitting/early-stopping etc. by myself?
- In a practitioner's point of view, $m$ must be found via grid-search on training/validation data. How susceptible are conventional models to overfitting?

Addressing these points would improve my assessment of the paper further.

**Ethical Concerns:**

["NO or VERY MINOR ethics concerns only"]

**Final Justification:**

The authors provide a point-by-point reply to my concerns and overall they are satisfactory.

**Limitations:**

The authors provided Limitations section at Appendix B, but highlighting several points e.g., Global Lipschitz constant assumption and non-convexity of minmax optimization would be beneficial for readers.

**Paper Formatting Concerns:**

N.A.

**Quality:**

4

**Strengths And Weaknesses:**

## Strengths.
- Theoretical advance over prior work. Extends the analysis to agnostic settings and relaxed ambiguity assumptions of Cheng et al. (2023) while still proving $O(n^{-1/2})$ convergence rates.
- Clear link between smoothness and effective interval width. The “reduced-interval lemma” (Thm 3.6) formalizes how a global Lipschitz constant $m$ shrinks the set of plausible labels, providing an intuitive knob for bias–variance trade-off.
- Two complementary risk surrogates. By analyzing both projection and min-max objectives under the same smoothness framework, the paper offers upper- and lower-risk envelopes that bracket the unknown true loss.
- Simple, architecture-agnostic implementation. Enforcing global Lipschitzness via spectral normalisation can be applied to any feed-forward network without architectural redesign.
- Comprehensive supplementary material. The appendix provides full proofs, additional datasets, hyperparameter grids, and ablation studies, facilitating in-depth verification.

## Weaknesses.
The paper was an enjoyable read, but I found the following potential weaknesses and questions. I am willing to reconsider my score once these issues are addressed.

### Potentially unclear link between the theory and experiments.
Fig. 7 is confusing to me in sense that the claimed smoothness-realizability tradeoff (p.8, l.268-272) appears only on one of the three datasets (abalone) shown; the center (airfoil) and right (concrete) plots show monotonic decrease, without showing a minima due to the tradeoff between the term (b) and Rademacher complexity. Please clarify and discuss whether it is expected from the theory, or limitation of it.

### Global Lipschitz constraint may be unrealistic.
The assumption that a single scalar $m$ upper bounds $|\nabla f|$ for all $x$ might be difficult to be satisfied in high-dimensional or heterogeneous data: it may have risks of e.g., underfitting sharp local variations or exploding the output scale when the data manifold is low-dimensional inside $\mathbb{R}^d$.

### Reproducibility concern.
Code is currently unavailable (though promised upon acceptance), preventing reviewers from assessing reproducibility.

### (Minor) Synthetic-only supervision.
It seems that all interval targets are generated by an artificial random procedure; no evaluation on naturally occurring interval or censored data is provided.

---

> ### Author Response · Authors · 2025-08-01
>
> Thank you for the thoughtful feedback. Please find our individual responses below.
>
> > Potentially unclear link between the theory and experiments.
> Fig. 7 is confusing to me in sense that the claimed smoothness-realizability tradeoff (p.8, l.268-272) appears only on one of the three datasets (abalone) shown; the center (airfoil) and right (concrete) plots show monotonic decrease, without showing a minima due to the tradeoff between the term (b) and Rademacher complexity. Please clarify and discuss whether it is expected from the theory, or limitation of it.
>
> Thank you for this question. Our theoretical upper bound consists of three main terms: (a) approximation error, (b) effective interval width, and (c) Rademacher complexity. While the reviewer correctly points to the trade-off between terms (b) and (c), our hypothesis is that for these specific datasets, the empirically dominant trade-off is actually between the approximation error (a) and the effective interval width (b), with the complexity term (c) playing a smaller role.
>
> 1. **The U-Shape :** The U-shape appears when, for a small Lipschitz constant m, the approximation error (a) is large but the effective interval width (b) is small. Conversely, as m grows, the approximation error (a) shrinks, but the model's reliance on the original supervisory intervals causes the effective width (b) to grow, eventually increasing the total error.
> 2. **The Monotonic Decrease:** The key insight is that term (b) is highly dependent on the "informativeness" of the initial supervisory intervals. On datasets like Airfoil and Concrete, we believe that the provided intervals are already very tight and informative. Consequently, even as m increases and the approximation error (a) drops, the effective interval width (b) remains small. The trade-off is muted because term (b) never grows large enough to cause an inflection in the error curve within our tested range.
>
> In summary, the shape of the curve is dataset-dependent and hinges on the quality of the interval supervision itself. We will add a discussion to the paper to clarify this nuance. In addition, we also have plots for 2 additional dataset in Appendix J (housing and power-plant dataset) where we observe the U-shape and Monotonic decrease plot respectively.
>
> > Global Lipschitz constraint may be unrealistic.
> The assumption that a single scalar m  upper bounds gradient f for all  might be difficult to be satisfied in high-dimensional or heterogeneous data: it may have risks of e.g., underfitting sharp local variations or exploding the output scale when the data manifold is low-dimensional inside .
>
> We agree with this limitation and have discussed this in the limited work section, we note that other similar assumptions, such as a modulus of continuity, could also lead to analogous results which would be an interesting direction for future work.
>
> > Reproducibility concern. Code is currently unavailable (though promised upon acceptance), preventing reviewers from assessing reproducibility.
>
> Unfortunately, we are strictly bound by the conference policy which prohibits sharing new materials or external links during the rebuttal period. This prevents us from providing the codebase at this stage.
> Given these constraints, we are more than happy to answer any specific questions about our implementation, model architecture, or hyperparameter settings to the best of our ability. We reaffirm our commitment to make the complete, documented codebase publicly available immediately upon acceptance.
>
> > In a practitioner's point of view,  m must be found via grid-search on training/validation data. How susceptible are conventional models to overfitting?
>
> We treat m as a hyperparameter tuned via standard validation. The effectiveness of this approach is shown in Table 4 (Appendix L), where our method consistently and significantly outperforms the MLP baseline, confirming it provides robust regularization rather than simply overfitting.

---

> > ### Comment · Reviewer_U3a1 · 2025-08-05
> >
> > Thank you for the comprehensive and thoughtful responses. Your clarifications satisfactorily address my concerns, so I have increased my overall score. I look forward to seeing the final version at NeurIPS and to experimenting with the released code.
> >
> > Minor comment: Table 4 shows a large degradation only on the GPU dataset. While this never be a weakness of the proposed approach, I'm curious if you have a hypothesis, e.g., interval width, feature scale, dimensionality, etc. that explains why the Lipschitz constraint hurts on this dataset?

---

### Official Review · Reviewer_oLAy · 2025-06-30

**Clarity:** 3
**Significance:** 2
**Originality:** 2
**Rating:** 4
**Confidence:** 3

**Summary:**

The paper studies regression with interval-valued labels under an m-Lipschitz hypothesis class, where each observed interval captures uncertainty about the true target.
When the true target is not available, it adopts two practical surrogates, an (optimistic) projection loss and a closed-form min–max loss, plus an auxiliary network for the inner maximization. This paper also provides a decomposed generalization bound on the loss.

**Questions:**

1. What does PL Mean mean in the experiment?

2. Could this approach be applied beyond tabular data? I think the application on the unstructured data, such as images, is very important and practical.

3. Is there a better way of approximating the Lipschitz constant? It seems the approximated ones in the experiment does not lead to optimal results.

**Ethical Concerns:**

["NO or VERY MINOR ethics concerns only"]

**Final Justification:**

I appreciate the theoretical foundation of the proposed work. The authors addressed my concern regarding noisy label learning techniques. Although the experimental evaluation could be improved, I believe the overall strength outweigh the weakness. Therefore I raised my score to 4.

**Limitations:**

yes

**Paper Formatting Concerns:**

No formatting concern observed.

**Quality:**

2

**Strengths And Weaknesses:**

**Strengths**

1. In general, the ideas are clearly conveyed.

2. The generalization bound decomposition indicates the usefulness of the proposed method.

3. The setting is interesting and practical in real-world data collection, where the labels are often noisy.

**Weaknesses**

1. While I appreciate the theoretical results, the technical contribution is unclear to me. The projection loss is very similar to prior works (such as [1]), while the minmax formulation seems to be a special or simplified case of robust ML (such as [2, 3]), where the feature variations/uncertainty do not exist.

2. The approach for minmax formulation that utilizes an auxiliary worst-case learner is not connected to interval targets.

3. Since the ultimate goal of this work is to reduce the overall loss of the classifier given interval targets, there are other baselines that could be considered in the evaluation. For instance, the evaluation could be strengthened by comparing with works that learn robust models from noisy labels, such as [4]. To construct their setting, the noisy labels could be sampled from the interval ranges.

[1] Cheng, Xin, Yuzhou Cao, Ximing Li, Bo An, and Lei Feng. "Weakly supervised regression with interval targets." In International Conference on Machine Learning, pp. 5428-5448. PMLR, 2023.
[2] Mirman, Matthew, Timon Gehr, and Martin Vechev. "Differentiable abstract interpretation for provably robust neural networks." In International Conference on Machine Learning, pp. 3578-3586. PMLR, 2018.
[3] Wang, Shiqi, Huan Zhang, Kaidi Xu, Xue Lin, Suman Jana, Cho-Jui Hsieh, and J. Zico Kolter. "Beta-crown: Efficient bound propagation with per-neuron split constraints for neural network robustness verification." Advances in neural information processing systems 34 (2021): 29909-29921.
[4] Kim, Chris Dongjoo, Sangwoo Moon, Jihwan Moon, Dongyeon Woo, and Gunhee Kim. "Sample selection via contrastive fragmentation for noisy label regression." Advances in Neural Information Processing Systems 37 (2024): 127561-127609.

---

> ### Author Response · Authors · 2025-08-01
>
> Thank you for the thoughtful feedback. Please find our individual responses below.
>
> > While I appreciate the theoretical results, the technical contribution is unclear to me. The projection loss is very similar to prior works (such as [1]), while the minmax formulation seems to be a special or simplified case of robust ML (such as [2, 3]), where the feature variations/uncertainty do not exist.
>
> We appreciate the reviewer's request for clarification on our novel contributions. We have summarized them below:
> 1. **Projection Loss**: We clarify that our work significantly generalizes the projection loss proposed in Cheng et al. [1]. Our key contributions are:
> - **Generalizes the Loss Function:** While [1] focuses specifically on the l1​ loss, our theoretical analysis is general and applies to a broader class of loss functions.
> - **Relaxes Strong Assumptions:** Our framework removes the restrictive assumptions of realizability and small ambiguity degree required by [1]. This makes our analysis more practical and applicable to a wider range of real-world, agnostic learning scenarios.
>
> 2. **Robust ML:** We appreciate the connection to robust ML. The fundamental difference is that robust methods like [2, 3] aim to enforce model stability against adversarial perturbations of the input features. In contrast, our work addresses a bounded (not arbitrary/adversarial) uncertainty in the output labels, since the true label must lie within the known interval.
>
>
>
>
>
> > The approach for minmax formulation that utilizes an auxiliary worst-case learner is not connected to interval targets.
>
>
> The interval targets are a core components of our min-max formulation since the worst-case labels are generated from a hypothesis that lies inside the interval targets.
>
>
> > Since the ultimate goal of this work is to reduce the overall loss of the classifier given interval targets, there are other baselines that could be considered in the evaluation. For instance, the evaluation could be strengthened by comparing with works that learn robust models from noisy labels, such as [4]. To construct their setting, the noisy labels could be sampled from the interval ranges.
>
> Noisy label regression would discard the interval information, and treat a fraction of the data as having arbitrary outliers, which is not the case in our setting. There may be interesting ways to leverage the noisy label literature, leading to a separate approach to the problem, but it is not obvious de facto what the right reduction is. Hence we do not feel it is a necessary baseline. We have also elaborated on this in the general response.
>
> > What does PL Mean mean in the experiment?
>
> We apologize for the confusion. PL Mean refers to Pseudo labels (mean) which is our proposed method for solving the minmax objective with constraints. We provided full definitions in Appendix G.
>
>
>
> > Could this approach be applied beyond tabular data? I think the application on the unstructured data, such as images, is very important and practical.
>
> Yes, it is also applicable to images or any other domains.
>
>
>
> > Is there a better way of approximating the Lipschitz constant? It seems the approximated ones in the experiment does not lead to optimal results.
>
> Our approach is a standard heuristic for approximating the Lipschitz constant from the dataset which is not necessarily optimal for the LipMLP.  In practice, one can just treat the Lipschitz constant for the LipMLP as a hyperparameter and tune them accordingly. This leads to a significant performance gain when compared with the standard MLP (see Table 4 in Appendix L).

---

> ### Comment · Reviewer_oLAy · 2025-08-04
>
> I appreciate the detailed response from the authors. I have some concerns remaining and would like to discuss more:
> - **Technical contribution of minmax formulation**. Technically, robust ML looks into $min_{M}\ max_{D\in D^P}\ loss(M, D)$ with $M$ and $D$ being the model and data, and $D$ might come from a set of possible datasets with bounded perturbations $D^P$. The difference between this work's minmax formulation and the formulation of most robust ML techniques is that, this work looks at bounded perturbation of labels, while previous works look at that of features. However, the inner optimization for feature perturbation is way more challenging, and can be easily adapted to find worst-case labels.
> - **Larger, unstructured data experiment**. This work closely connects to partial label learning, and most works in that domain have been tested on larger, unstructured data (such as some early works [1,2]). Therefore, I believe the evaluation of this work could be strengthened by including such data as well.
> - **Comparing with noisy label learning**.
>   - I understand that the settings of noisy label learning do not include a bounded interval target assumption. However, I view the interval bounds as a more restricted setting, and noisy label learning methods should be able to be applied here. For instance, we can use the mean of each interval (or a randomly sampled value inside the interval) as the noisy (pseudo) label, and apply the noisy label learning techniques.
>   - A more interesting question is, whether the pseudo-labeling based on the min-max formulation proposed in this work could be combined with subsequent noisy label learning and show better performance.
>
>
> [1] Lv, Jiaqi, Miao Xu, Lei Feng, Gang Niu, Xin Geng, and Masashi Sugiyama. "Progressive identification of true labels for partial-label learning." In international conference on machine learning, pp. 6500-6510. PMLR, 2020.
>
> [2] Feng, Lei, Jiaqi Lv, Bo Han, Miao Xu, Gang Niu, Xin Geng, Bo An, and Masashi Sugiyama. "Provably consistent partial-label learning." Advances in neural information processing systems 33 (2020): 10948-10960.

---

> > ### Author Response · Authors · 2025-08-04
> >
> > Thank you for your follow-up questions. We've provided our responses below.
> >
> > > Technical contribution of minmax formulation
> >
> > We agree that our work and prior work in robust ML both follow the min-max formulation that the reviewer suggested. In most work on robust ML, the goal is to find the feature perturbation that maximizes the loss, while in our setting, we find the label that maximizes the loss. However, regarding the inner optimization, our setting is much simpler to solve. In particular, for any given model, a closed-form solution for the worst-case label exists. Intuitively, this is simply the label that is as far as possible from the model’s prediction while remaining valid (i.e., lying inside the interval). This leaves only two candidate points: the two endpoints of the interval. On the other hand, in robust ML, one wants to find the features that maximize the loss, which is a much more difficult optimization problem.
> >
> > Furthermore, we also explore the non-trivial consequences of this formulation. As shown in our paper, naively training on these worst-case labels can lead to suboptimal performance (Proposition 5.4). We address this by incorporating a smoothness property by only allowing the smooth worst-case labels which we demonstrate improves results.
> >
> > > Larger, unstructured data experiment.
> >
> > We thank the reviewer for this constructive suggestion. Our current evaluation provides compelling evidence of our method's robustness and generalizability. To this end, we conducted extensive experiments on 18 additional diverse tabular datasets from the benchmark by Grinsztajn et al. [1]. As shown in Appendix L (Table 4), **our LipMLP significantly outperforms the standard MLP with projection loss (baseline) on 14 of these 18 datasets** and are on par with the rest, demonstrating its broad effectiveness.
> >
> > Furthermore, these strong results on tabular data provide a robust indication of our method's potential on unstructured data. Our main baseline, an MLP with projection loss, is equivalent to the LM loss used in Cheng et al. [2]. This prior work has already demonstrated this baseline's effectiveness on unstructured data, including computer vision datasets (AgeDB, IMDB-WIKI) and an NLP dataset (STS-B). The fact that our LipMLP consistently outperforms this baseline strongly suggests its potential for competitive performance in unstructured settings as well. While we leave a direct verification to future work, we agree this is a promising direction and believe our results provide a strong foundation for it.
> >
> >
> > [1] Grinsztajn, Léo, Edouard Oyallon, and Gaël Varoquaux. "Why do tree-based models still outperform deep learning on typical tabular data?." Advances in neural information processing systems 35 (2022): 507-520.
> >
> > [2] Cheng, Xin, Yuzhou Cao, Ximing Li, Bo An, and Lei Feng. "Weakly supervised regression with interval targets." In International Conference on Machine Learning, pp. 5428-5448. PMLR, 2023.
> >
> > > Comparing with noisy label learning.
> >
> > We thank the reviewer for this insightful comparison. We respectfully argue that the interval target setting is not a more restricted case of noisy label learning, but rather a distinct problem formulation that preserves crucial information. The key difference is that our setting explicitly leverages the hard constraints provided by the interval boundaries.
> > To illustrate why this distinction is critical, let's consider the proposed reduction strategy. Assume the true target function is constant, and we observe two intervals: [−2000,1] and [−1,1000].
> >
> > 1. **Interval targets:** By using the interval information directly, we know the solution must lie in the intersection of all intervals. This immediately constrains the target value to the narrow range of [−1,1], a highly informative signal.
> >
> > 2. **Noisy Label Reduction:** Converting these intervals to pseudo-labels via their midpoints (−999.5 and 499.5) destroys this critical boundary information. Any subsequent noisy label learning technique would see two extremely noisy labels and would struggle to recover the simple underlying function.
> >
> > This example highlights that converting intervals to single noisy labels can discard the most valuable information in the data. For this reason, we believe methods tailored to the interval structure are necessary. However, we agree with the reviewer’s more nuanced point: exploring how our min-max formulation could be combined with noisy label techniques e.g. how to learn from noisy intervals, is indeed an interesting and promising direction for future work.

---

> > > ### Comment · Reviewer_oLAy · 2025-08-05
> > >
> > > I thank authors for the detailed clarifications. I raised my score and I have no further questions.

---

### Official Review · Reviewer_1WsC · 2025-06-30

**Clarity:** 3
**Significance:** 3
**Originality:** 3
**Rating:** 5
**Confidence:** 2

**Summary:**

This paper studies regression with interval targets, where the true labels are only known to lie within given intervals rather than being exact values. To address this setting, the authors propose two learning strategies: (1) minimizing a projection loss that penalizes predictions falling outside the intervals, and (2) a min-max objective that minimizes the worst-case loss over possible target values within each interval. They develop novel generalization bounds for the projection-based method under both realizable and agnostic settings, removing restrictive assumptions from prior work by leveraging the smoothness (e.g., Lipschitz continuity) of the hypothesis class. Theoretical insights reveal that smoother models implicitly reduce the effective size of interval constraints, improving predictive performance. Empirical results on real-world datasets support the theory and demonstrate that both proposed methods achieve state-of-the-art accuracy in interval-based regression tasks.

**Questions:**

Can a similar generalization error bound be established for the min-max objective method? If so, under what conditions would it match or differ from the projection-based analysis?

**Ethical Concerns:**

["NO or VERY MINOR ethics concerns only"]

**Final Justification:**

Thanks for the response. I will maintain my score.

**Limitations:**

See weakness

**Quality:**

3

**Strengths And Weaknesses:**

Strength
1. The paper is clearly written, and the theoretical results appear to be rigorous and well-justified.
2. The work significantly improves upon prior results by removing the restrictive realizability and small ambiguity degree assumptions, instead establishing generalization guarantees under a smoothness condition on the hypothesis class. The empirical results are comprehensive and demonstrate that the proposed methods achieve strong performance across multiple real-world datasets.
3. The empirical results are comprehensive and demonstrate that the proposed methods achieve strong performance across a few real-world datasets.

Weakness
It would strengthen the paper to include a discussion on the tightness of the provided generalization error bounds. In particular, can lower bounds be established to show whether the derived rates are optimal or near-optimal?

---

> ### Author Response · Authors · 2025-08-01
>
> Thank you for the thoughtful feedback. Please find our individual responses below.
>
>
>
> > It would strengthen the paper to include a discussion on the tightness of the provided generalization error bounds. In particular, can lower bounds be established to show whether the derived rates are optimal or near-optimal?
>
> Our bound can be tight for a certain hypothesis class, for example, a class of constant hypotheses. We discussed this in Appendix E (e.g. see eq (88)).
>
>
>
> > Can a similar generalization error bound be established for the min-max objective method? If so, under what conditions would it match or differ from the projection-based analysis?
>
>
> Yes, a finite-sample generalization bound for the min-max objective can be derived from our population-risk analysis in Eq. (23) using standard uniform convergence techniques. This bound would differ from the projection-based analysis because the min-max loss is, by definition, a pessimistic upper bound on the true loss, while the projection loss is an optimistic surrogate. Consequently, a generalization bound on the min-max risk will be inherently looser, as it provides a more conservative, worst-case guarantee on performance.

---

> > ### Comment · Reviewer_1WsC · 2025-08-05
> >
> > Thanks for the response. I will maintain my score.

---

### Author Response · Authors · 2025-08-01
**Overall response**

We sincerely thank all reviewers for their time and insightful feedback. We are grateful for the following strengths they identified in our work:

1. **Theoretical Contribution:** Our work was described as a **“Theoretical advance over prior work”** (U3a1) that **“significantly improves upon prior results”** (1WsC) by relaxing restrictive assumptions.

2. **Practical Implementation:** The proposed algorithm was noted for being **“practical... easy to implement”** (N93J) and having a **“simple, architecture-agnostic implementation”** (U3a1).

3. **Empirical Validation:** Our experiments were found to be **“comprehensive”** (1WsC) and demonstrated that the proposed method is **“empirically effective”**(N93J) and has a **“strong performance”**(1WsC).

4. **Clarity:** The paper was praised for its presentation, with ideas that are **“clearly conveyed”** (oLAy) and theory that is **“thorough and well-presented”** (N93J), **“rigorous and well-justified”** (1WsC).


The main concern raised revolves around a desire to see the comparison of our framework to the setting of regression with noisy labels (oLAy, N93J). While being conceptually related, we believe there are fundamental differences in the problem formulation that make a direct comparison challenging.


**On the Connection to Regression with Noisy Labels:**

1. **Information Structure:** Reducing an interval to a single "noisy" point discards the valuable structural information on the upper and lower bounds of the true label.
2. **Difference in "Noise" Models:** The assumptions in the two fields are quite distinct. Learning with Noisy Labels typically handles arbitrary outlier noise in a fraction of the data. On the other hand, in our setting, all labels are uncertain but are guaranteed to lie within a known interval.

There may be interesting ways to leverage the noisy label literature, leading to a separate approach to the problem, but it is not obvious de facto what the right reduction is. Hence we do not feel it is a strong connection or a necessary baseline.

We will address other suggestions and questions in individual responses. Thank you again for your hard work and consideration!

---

### Author Response · Authors · 2025-08-02

Dear Reviewers and Area Chair,

We sincerely apologize for submitting our response after the official deadline. This was due to a genuine oversight on our part, and we take full responsibility for the error.

We understand that this is outside the standard process and will completely respect your decision should you choose not to consider these comments.

Thank you for your time and for the valuable feedback on our work.

---

### Note · Authors · 2025-08-12

We sincerely thank the reviewers and Area Chair for the valuable and insightful review process.

The discussion period was particularly helpful. We are grateful that our clarifications were well-received and helped resolve initial concerns. We are committed to incorporating all remaining feedback on the paper's clarity and organization into the final version.

Encouraged by the reviewers' interest, we are also preparing our code for public release. We believe the paper has been greatly strengthened by this process and respectfully await the committee's decision.

---

### Decision · Program_Chairs · 2025-09-17

**Decision:**

Accept (poster)

**Comment:**

The paper provides learning bounds for supervised learning with interval targets. While previous bounds are limited to the realizable case with small ambiguity degree assumption, this paper presents bounds in the agnostic case with smooth function classes. The bound is insightful with terms that capture tradeoff between approximation error and smoothness, apart from usual estimation error that decays as 1/\sqrt{n}. Learning under worst case labels, using minmax obj, is also studied.

The paper is written well, easy to follow, with good organization of content. The theoretical results are interesting considering the new setting of interval targets. The analysis kind of exhaustive complementing gaps in literature. The reviews also agree with these points broadly after rebuttal discussions. Hence the work is ready for publication.